# FOCUS ON THIS, NOT THAT! STEERING LLMS WITH ADAPTIVE FEATURE SPECIFICATION

## ABSTRACT

Despite the success of Instruction Tuning (IT) in training large language models (LLMs) to perform arbitrary user-specified tasks, these models often still leverage spurious or biased features learned from their training data, leading to undesired behaviours when deploying them in new contexts. In this work, we introduce *Focus Instruction Tuning* (FIT), which trains LLMs to condition their responses by "focusing on" specific features whilst ignoring others, leading to different behaviours based on what features are specified. Across several experimental settings, we show that focus-tuned models can be adaptively steered by focusing on different features at inference-time: for instance, robustness can be improved by focusing on task-causal features and ignoring spurious features, and social bias can be mitigated by ignoring demographic categories. Furthermore, FIT can steer behaviour in new contexts, generalising under distribution shift and to new unseen features at inference time, and thereby facilitating more robust, fair, and controllable LLM applications in real-world environments.

## 1 INTRODUCTION

Instruction Tuning (IT) (Zhang et al., 2023), a specialised form of supervised fine-tuning (SFT), has become an essential step in the process of developing effective instruction-following large language models (LLMs) (Ouyang et al., 2022; Touvron et al., 2023; Chen et al., 2024). While extensive pre-training to perform next token prediction allows LLMs to extract common patterns and knowledge from large text corpora, IT fine-tunes these models on input-output pairs complemented by natural-language task instructions, teaching them to perform open-ended language-based tasks given instructions (Huang et al., 2023). However, despite the improvements observed in zero-shot generalisation from IT, recent studies suggest that some of these gains may be superficial, stemming from the models' ability to learn task template formats or spurious input/output correlations rather than a more generalisable instruction-following capability (Kung & Peng, 2023; Ghosh et al., 2024). As a result, LLMs may fail to generalise to new contexts where the same templates or spurious correlations are not present (Kung & Peng, 2023).

To address these limitations, we propose *Focus Instruction Tuning* (FIT), an extension of traditional IT where LLMs are also fine-tuned with respect to an instruction regarding which features to "focus on" or "not focus on". Training LLMs to condition responses on the provided focus specification, where responses to the same input differ based on the specified features, allows end users to dynamically steer model behaviour by indicating which features should play a role or be ignored in performing a task. For instance, in Figure 1, we show how FIT can be used to steer a model to ignore gender stereotypes and focus instead on task-relevant information in order to correctly solve a question-answering task. In our experiments, we demonstrate that FIT is effective at steering models to *ignore known spurious features* while *focusing on causal (task-relevant) features* (see Section 4) across a variety of features and tasks such as sentiment analysis and natural language inference, and can be used to mitigate social bias by ignoring demographic features in question-answering. We find that FIT is also robust to distribution shifts over feature values, and can generalise to new, held-out features not encountered during training. In sum, our primary contributions are as follows:

1. We introduce *Focus Instruction Tuning* (FIT), a method that allows users to flexibly and dynamically specify what features a model should or should not focus on when performing a task at inference time. FIT enables practitioners to incorporate domain-specific knowl-

Figure 1: **Focus Instruction Tuning (FIT).** In the example above, a model that is simply Instruction Tuned would follow biases present in the training set. On the other hand, a FIT model can dynamically focus or ignore specific features of the input. Here we report a sample from BBQ (Parrish et al., 2022) (a question- answering dataset designed to elicit incorrect answers from models that perform the task by resorting to socially biased stereotypes). Both a man and a woman are present at an engineering candidate's interview. When asked who made the technical question, due to the pre-training and instruction tuning biases, a model could respond the man did, despite the conflicting evidence present in the context. A focus-tuned model can ignore the gender feature and focus instead on the interview content.

edge about causal, spurious, or bias-relevant features in order to steer models according to the desired feature specification.

2. We experiment with FIT across several key NLP tasks, including sentiment analysis, natural language inference, and question-answering. We find that FIT is highly effective for steering behaviour on all tasks with respect to a variety of lexical, distributional, semantic, and demographic features.

3. We show that FIT generalises with respect to (a) distribution shift over feature values and (b) focusing on new features not seen during training.

## 2 BACKGROUND AND RELATED WORK

### 2.1 SPURIOUS FEATURE LEARNING

Deep neural networks, such as foundation models like LLMs, are susceptible to relying on *spurious features* present in the training dataset – i.e., input features that are correlated with outputs in the training distribution, but are not correlated in all test distributions (Ye et al., 2024). Relying on spurious features leads models to fail to generalise under distribution shifts where such correlations may no longer hold Wang et al. (2023a). Spurious features have been extensively studied in computer vision, encompassing features such as background colour (Xiao et al., 2021; Venkataramani et al., 2024; Arjovsky et al., 2019) or texture (Geirhos et al., 2018; Baker et al., 2018), and are also prevalent in many widely used NLP benchmarks (Sun et al., 2024b; Borkan et al., 2019). For instance, the token SPIELBERG is spuriously correlated with positive sentiment in datasets like SST-2 (Socher et al., 2013b), meaning that models trained on SST-2 may learn to predict sentiment by leveraging these spurious features instead of more general sentiment features (Wang & Culotta, 2020). This reliance on non-causal features undermines the robustness of models in generalising to distribution shift.

A variety of approaches have been explored to detect and mitigate the effects of spurious feature learning, particularly under distribution shifts. Traditional approaches include prompt engineering (Sun et al., 2024b), regularisation techniques (Arjovsky et al., 2019; Chew et al., 2024), and directly incorporating causal inference strategies (Wang & Culotta, 2020; 2021; Udomcharoenchaikit et al., 2022). Substantial work in mechanistic interpretability has also aimed to discover models' representation and use of task-causal or spurious features: for instance, causal probing (which trains probing classifiers to recognise and modify supervised feature representations encoded by foundation models; see Belinkov, 2022; Canby et al., 2024; Davies & Khakzar, 2024) has been used to study how models leverage causal versus spurious features features in the context of a given task (Ravfogel et al., 2021; Lasri et al., 2022; Davies et al., 2023). Other works have leveraged unsupervised mechanistic interpretability methods, such as circuit discovery techniques (Wang et al., 2023b; Conmy et al., 2023) and sparse auto-encoders (Subramanian et al., 2018; Yun et al., 2021), to im-

prove generalisation by discovering spurious features leveraged by models in performing a given task and ablating their use of these features (Gandelsman et al., 2024; Marks et al., 2024). Finally, concept removal methods locate and manipulate supervised feature representations corresponding to bias features encoded by foundation models in order to remove these features (Ravfogel et al., 2020; 2022; 2023; Iskander et al., 2023; Belrose et al., 2024; Kuzmin et al., 2024).

## 2.2 CONTROLLING LLMS

**Instruction Tuning.** Due to the next-word prediction training objective, large language models (LLMs) often struggle by default to generate outputs that align with human instructions in downstream applications (Huang et al., 2023). Instruction-tuning (IT) mitigates this issue by fine-tuning pre-trained LLMs on datasets composed of instruction-response pairs (Zhang et al., 2023), aiming to align the responses of the fine-tuned model more closely with the distributions preferred by humans (Ouyang et al., 2022). There are several popular approaches for collecting IT training data, such as using human-annotated data (Dolly, 2023), extracting datasets from existing collections (Longpre et al., 2023; Mishra et al., 2022), or gathering data from internet sources (Zhou et al., 2024). IT datasets can also be synthesised with LLMs, either by bootstraping them from the same model that will be instruction-tuned on them (Wang et al., 2023c; Chen et al., 2024), or by distilling from a larger or more powerful model to instruction-tune smaller models (Taori et al., 2023; Mitra et al., 2023; Xu et al., 2023).

Despite the success of IT in zero-shot generalisation, Gudibande et al. (2023) find that improvements on many downstream benchmark tasks may be largely due to coverage of task data within IT training datasets; and bootstrapping IT methods (which, in principle, might not be subject to this issue provided they synthesise novel IT task instances) require a robust and effective LLM for fine-tuning to avoid degenerate training cycles (Zhang et al., 2023). Furthermore, Kung & Peng (2023) show that some of the downstream performance gains from IT can be attributed to models' ability to learn surface-level patterns, such as the required answer format, rather than acquiring more generalisable instruction-following skills. These limitations underscore the need for advancements in supervised fine-tuning (SFT) methods beyond IT to facilitate more predictable and reliable control of downstream model behaviours.

**Refocusing LLMs.** Several methods have been proposed to better control instruction-tuned models both during and after training. Llama-Guard (Inan et al., 2023) fine-tunes LLMs to detect predefined risk features in inputs and outputs based on a user-specified taxonomy, such as identifying sexual content in inappropriate contexts. JsonTuning (Gao et al., 2023) enhances traditional instruction tuning by enforcing structured input and output formats in JSON, clarifying task requirements and reducing sensitivity to paraphrasing (Sun et al., 2024a). In contrast, Focus Instruction Tuning (FIT), as introduced in this work, provides a more flexible and powerful approach. While Llama-Guard operates only post-training and is limited to the safety domain, FIT enables fine-grained control both during and after training, conditioning models on a broader range of features across arbitrary domains via natural-language specifications. Moreover, unlike JsonTuning, which is restricted to enforcing output structure, FIT allows users to specify input features, enabling the model to ignore spurious correlations or highlight task-relevant attributes.

## 3 METHODOLOGY

**Preliminaries.** We consider a pre-trained, decoder-only large language model (LLM) $p_\theta$ that models the probability distribution over its vocabulary $\mathcal{V}$ autoregressively. For an input sequence $x = [x_1, \ldots, x_L] \in \mathcal{V}^L$, the joint probability of $x$ is given by $p_\theta(x) = \prod_{i=1}^{L} p_\theta(x_i|x_{<i})$, with $p_\theta(x_1|\emptyset) = p_\theta(x_1)$. In traditional supervised learning, for a sample $(x, y) \sim \mathcal{D}$, the conditional likelihood of the output $y$ given input $x$ is $p_\theta(y|x) = \prod_{i=1}^{L} p_\theta(y_i|x, y_{<i})$, with $p_\theta(y_1|\emptyset) = p_\theta(y_1)$; and in supervised fine-tuning (SFT) of LLMs, this manifests as minimising the negative log-likelihood (NLL) of $y$ given $x$.

In instruction tuning (IT) (Zhang et al., 2023), a form of SFT, an additional task instruction $I$ accompanies the input-output pair $(x, y)$, forming a tuple $(I, x, y)$. The objective becomes minimising the NLL of $y$ given both $I$ and $x$, i.e., $p_\theta(y|I, x)$.

**Focus Instruction Tuning (FIT).** We introduce Focus Instruction Tuning (FIT), a specialised form of instruction tuning that trains LLMs to adjust their responses based on user-specified features provided in natural language.

Let $\mathcal{F}$ denote the set of possible features (e.g., specific keywords, sentiment, verb tense, demographic information, etc.) that the model can be instructed to focus on or ignore when generating responses. We consider a set of natural language instructions to focus or rule out specified features in $\mathcal{F}$ which we term the focus instruction set $\mathcal{I}_{\text{focus}}$.[1] Explicitly, we define $\mathcal{I}_{\text{focus}}$ as

$$\mathcal{I}_{\text{focus}} = \{\emptyset, \ \text{focus}(F_i), \ \text{ignore}(F_j), \ \text{focus}(F_i) \wedge \text{ignore}(F_j) \mid F_i, F_j \in \mathcal{F}\}, \tag{1}$$

where: $\emptyset$ denotes an empty focus instruction with no features to focus on or to ignore; $\text{focus}(F_i)$ is an instruction to focus on feature $F_i$; $\text{ignore}(F_j)$ is an instruction to ignore feature $F_j$; and $\text{focus}(F_i) \wedge \text{ignore}(F_j)$ is an instruction to focus on $F_i$ **whilst ignoring** feature $F_j$. We include the default prompt in order to aid the model in learning the underlying task as well as the ability to refocus its attention on specified features during FIT.

Consider a sample $(x, y) \sim p_{\text{data}}(x, y)$ drawn from an underlying data distribution and a focus instruction $I_{\text{focus}}$ drawn from a distribution $p_{\mathcal{I}_{\text{focus}}}$ over the set of focus instructions $\mathcal{I}_{\text{focus}}$. Then the likelihood of response $y$ conditioned on input $x$, task instruction $I$ (as in standard IT), and focus-instruction $I_{\text{focus}}$ is modelled as $p_\theta(y|I, I_{\text{focus}}, x)$.

**FIT Training.** Consider a classification task[2] with finite label space $\mathcal{Y}$, where a single *causal feature* $C \in \mathcal{F}$ is fully predictive of label $y \in \mathcal{Y}$ given input $x$ at both training time and under distribution shift (Koh et al., 2021). We also consider *spurious features* $S \in \mathcal{S} \subseteq \mathcal{F}$ from a *subset of spurious features* $\mathcal{S}$, where feature values[3] $s \in \text{Image}(S)$ for some spurious feature $S \in \mathcal{S}$ correlate with a label $y_s \in \mathcal{Y}$, where this correlation may change under distribution shift (Ming et al., 2022). Finally, we define $\mathcal{F}$ as the set of

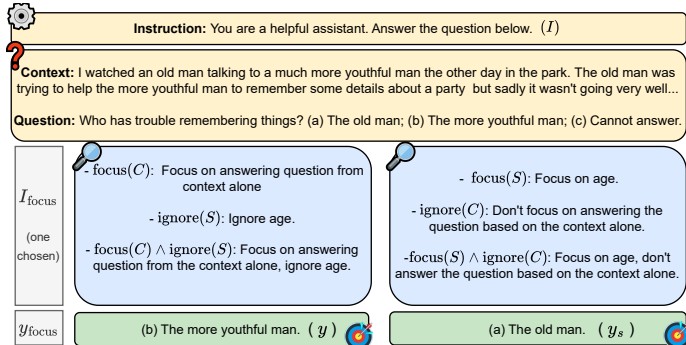

Figure 2: **Example of focus labels.** Focus labels for a modified example from BBQ. Here, age is a spurious feature.

features that may be included in focus instructions during training, consisting of the causal feature and the set of spurious features $\mathcal{F} = \{C\} \cup \mathcal{S}$.

For a sample $(x, y) \sim p_{\text{data}}(x, y)$, we specify the *focus label* $y_{\text{focus}} = y_{\text{focus}}(x, y, I_{\text{focus}}) \in \mathcal{Y}$ that depends on the ground truth label $y$ and focus instruction $I_{\text{focus}} \in \mathcal{I}_{\text{focus}}$. Intuitively, we define focus label $y_{\text{focus}}$ as $y_{\text{focus}} = y$ when either no focus features are specified (i.e., using the empty focus instruction), when the focus is on the underlying causal feature $C$, or when ignoring a spurious feature $S$; but when either the focus is on a spurious feature or the causal feature is ignored, $y_{\text{focus}}$ is defined as the label spuriously correlated with a particular value of the spurious feature present in input $x$. This changing target $y_{\text{focus}}$ trains the model to learn to adjust its responses based on specified features. Formally, we define $y_{\text{focus}}$ as:

$$y_{\text{focus}} = \begin{cases} y & \text{if } I_{\text{focus}} \in \{\emptyset, \text{focus}(C), \text{focus}(C) \wedge \text{ignore}(S), \text{ignore}(S) \mid S \in \mathcal{S}\}; \\ y_s & \text{if } I_{\text{focus}} \in \{\text{focus}(S), \text{focus}(S) \wedge \text{ignore}(F_j) \mid F_j \in \mathcal{F} \setminus \{S\}\}, \text{ for } s \in x; \\ y_s & \text{if } I_{\text{focus}} \in \{\text{ignore}(C)\} \text{ for sampled feature } S \in \mathcal{S} \text{ with value } s \in x. \end{cases} \tag{2}$$

---

[1]Examples of focus instructions specified in natural language include: "Make sentiment the central factor in your decision" and "Base your prediction solely on the presence of keywords. Exclude the logical relationship between the premise and hypothesis".

[2]For simplicity, we focus on classification here; but FIT is also applicable to generative tasks, as we show in our question-answering experiments.

[3]Note that we use uppercase to denote features, and lowercase to denote specific values of features.

where we again use $s \in x$ to denote the presence of the feature value $s$ in input $x$. See Figure 2 for a concrete example showing the focus label values for an example from the MNLI dataset under different focus instructions.

The objective of FIT training is to minimise the negative log-likelihood (NLL) of the response $y_{\text{focus}}$ conditioned on $I, I_{\text{focus}}, x$. Formally, we define the FT loss objective as:

$$\min_{\theta} \mathbb{E}_{(x,y) \sim p_{\text{data}}(x,y), \, I_{\text{focus}} \sim p_{\mathcal{I}_{\text{focus}}}(\mathcal{I}_{\text{focus}})} \left[ -\log p_{\theta} \left( y_{\text{focus}} \mid I, I_{\text{focus}}, x \right) \right]. \tag{3}$$

We define $p_{\mathcal{I}_{\text{focus}}}(\mathcal{I}_{\text{focus}})$ by placing a small probability mass on the empty focus instruction prompt $\emptyset$ in order to aid in learning the underlying task, and then uniformly distribute the remaining probability mass over the remaining non-empty feature instructions. The objective in Equation (3) can be optimised through sampling using stochastic gradient descent (SGD) with popular optimisers such as AdamW (Loshchilov & Hutter, 2019). Further details on FT optimisation are provided in Appendix A.1.

**Evaluating FIT under spurious correlations.** After introducing FIT above, we now turn to settings where we can empirically train and evaluate it. A key aspect of our evaluation is the use of known spurious correlations, which simulate real-world scenarios where models can be misled by features that are spuriously predictive of the output label. By adjusting the co-occurrence rate between spurious features and their associated labels, we can test FIT's ability to dynamically steer a model's responses depending on the features on which it is focusing or ignoring.

We define the co-ocurrence rate, or predictivity (Hermann et al., 2024), between spurious feature values and the label with which they are spuriously correlated by $\rho_{\text{spurious}}$. Specifically:

**Definition 1.** *(Defining $\rho_{spurious}$). Let $S \in \mathcal{S} \subseteq \mathcal{F}$ denote a spurious feature. Suppose that a value of $S$, say $s$, is spuriously correlated with label $y_s$. Then we define $\rho_{spurious}(s)$ as*

$$\rho_{spurious}(s) = \mathbb{P}(Y = y_s | X, S = s, s \in X) \tag{4}$$

*for some dataset sample $(x, y) \in \mathcal{D}$, where $S \in X$ denotes the presence of feature $S$ in example $X$.*

By varying $\rho_{\text{spurious}}(s)$, we can control the predictivity of spurious features and observe the model's behaviour when focusing on or ignoring these features as well as causal features.

Given a task with $N$ classes, we require $\rho_{\text{spurious}} = 1/N$ *within the training set*, ensuring that the underlying label distribution, $p(y|I, I_{\text{focus}}, x)$, is of maximum entropy when focusing on spurious features. This allows the model to better distinguish between causal and spurious features, as effectively minimising Equation (3) would require the model to make predictions without relying on the underlying causal feature when its attention is specified to focus on on spurious features. A more detailed exploration of this setting of $\rho_{\text{spurious}}$ during training can be found in Appendix A.1.

Next, we evaluate FIT across several test sets that capture different conditions of spurious correlations and distribution shifts:

- $\mathcal{D}_{\text{iid}}$: Held-out test samples with the same $\rho_{\text{spurious}}$ as in the training set.
- $\mathcal{D}_{\text{high}}$: Test samples with a higher $\rho_{\text{spurious}}$ than in the training set.
- $\mathcal{D}_{\text{low}}$: Test samples with a lower $\rho_{\text{spurious}}$ than in the training set.
- $\mathcal{D}_{\text{flipped}}$: Test samples where spurious feature values are flipped to co-occur with different labels than in the training set, with the same high $\rho_{\text{spurious}}$ as in $\mathcal{D}_{\text{high}}$.

We further evaluate FIT under distribution shifts, where the specific values taken by spurious features do not overlap between the training and test sets, by introducing one additional test set:

- $\mathcal{D}_{\text{shift}}(\rho_{\text{spurious}})$: Test datasets where the spurious feature values are distinct from those within the training set.

We evaluate over these datasets specifically on our SMNLI datset (c.f. Section 4.2).

## 4 EXPERIMENTS

In this section we empirically validate the effectiveness of FIT across a range of popular LLMs of varying sizes and on different NLP datasets, including classification and multi-choice question-answering tasks.

Before reporting the main results, we introduce the evaluation metric (focus accuracy) that we report, baselines, models, and training settings used throughout the experiments. In Section 4.1, we first verify that FIT performs well on the simpler SS dataset, a synthetic sentiment analysis dataset derived from SST-5 (Socher et al., 2013b). We then demonstrate in Section 4.2 that FIT generalises to more complex features and handles distribution shifts on the SMNLI dataset, a sub-sampled version of the MNLI dataset (Williams et al., 2018). Finally, in Section 4.3, we show that FIT has practical, real-world impact by effectively mitigating bias in the BBQ dataset (Parrish et al., 2022), where we further illustrate FIT's ability to generalise to new features seen for the first time when performing inference.

**Metrics.** We define the *focus accuracy* for a focus instruction $I_{\text{focus}} \in \mathcal{I}_{\text{focus}}$ as the proportion of samples where the model's prediction aligns with the focus label, $y_{\text{focus}}$, as specified in Equation (2). Specifically, for each sample $(x, y) \in \mathcal{D}$, the model produces a prediction $\hat{y} \sim p_\theta(y \mid I, I_{\text{focus}}, x)$ based on a fixed focus instruction $I_{\text{focus}} \in \mathcal{I}_{\text{focus}}$. The focus label, $y_{\text{focus}} = y_{\text{focus}}(x, y, I_{\text{focus}})$, corresponds to the target output given the focus instruction for the input $x$ with ground truth label $y$. Focus accuracy for focus instruction $I_{\text{focus}}$, denoted $\mathcal{A}_{\text{focus}}(I_{\text{focus}})$, is computed as the fraction of correct predictions with respect to the focus label:

$$\mathcal{A}_{\text{focus}}(I_{\text{focus}}) = \frac{1}{|\mathcal{D}|} \sum_{(x,y) \in \mathcal{D}} \mathbf{1}(\hat{y} = y_{\text{focus}}), \tag{5}$$

where $\mathbf{1}(\hat{y} = y_{\text{focus}})$ is the indicator function that equals 1 if the model's prediction $\hat{y}$ matches the focus label $y_{\text{focus}}$, and 0 otherwise.

We report focus accuracy for each model on all dataset splits, using the prompt types and focus instructions detailed in Appendix A.3. Generations are evaluated through simple pattern matching due to the use of constrained beam decoding. Further details are provided in Appendix A.2.

**Models and training settings.** We evaluate FIT using three popular LLMs that span a range of model sizes: `Llama-3.1-8B-Instruct` (Dubey et al., 2024), `Mistral-7B-Instruct-v0.3` (Jiang et al., 2023), and `Vicuna-13B-v1.5` (Chiang et al., 2023). The models are fine-tuned using parameter-efficient SFT with LoRA (Hu et al., 2021), leveraging Hugging Face's `SFTTrainer` (Wolf et al., 2020) with default hyperparameters. Early stopping is applied based on validation loss, as defined in Equation (3). For generation, we use constrained beam decoding (Anderson et al., 2017) and use fully verbalised (natural language) labels during both training and testing, except for the multi-choice BBQ dataset. For further training details, refer to Appendix A.1.

**Baselines.** We compare against in the main section of the paper: a few-shot baseline (Manikandan et al., 2023) and a SFT baseline. The SFT baseline, SFT($y_{\text{focus}}$), follows the same setup as the FIT method (trained on sampled inputs and focus labels), but without the inclusion of focus instructions during training. This ensures a fair comparison between FIT and the baseline, as both methods are trained on the same examples and labels (i.e., focus labels $y_{\text{focus}}$), with the only difference being the inclusion of focus instructions in FIT. This setup allows us to isolate and evaluate the specific impact of incorporating focus instructions in FIT. The few-shot baseline involves using 5 in-context examples uniformly sampled at random from the training set for each test example, where we use the same focus instruction for each in-context sample as for the test sample. In Appendix A.5, we include two additional baselines: zero-shot and vanilla SFT for a more complete comparison with FIT. Further details of baselines can be found in Appendix A.4.

### 4.1 VALIDATION OF FIT ON THE SS DATASET

**Spurious Sentiment dataset (SS).** We first evaluate FIT on a synthetic binary sentiment analysis dataset. Starting with SST-5 (Socher et al., 2013a), a 5-class sentiment analysis dataset, we use `Llama-3.1-70B-Instruct` (Dubey et al., 2024) to inject the spurious keywords *Pineapple* and *Bayesian* into all dataset examples in a natural way.[4] In this process, we preserve the original sentiment of the dataset examples and combine categories of positive and negative labels into single

---

[4]We observe that the LLM makes minimal changes to each input, sometimes only inserting the keyword where appropriate, and in other cases only adding a few words to create a more appropriate context (e.g., prepending "According to our Bayesian analysis," to a declarative clause). See Appendix A.7 for further details.

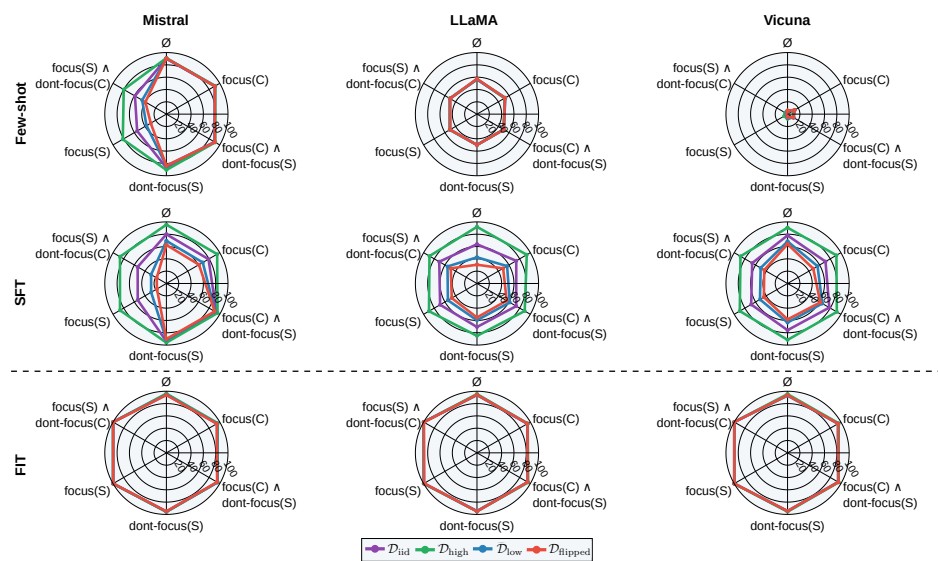

Figure 3: **SS focus accuracies** ($\uparrow$). Focus accuracy ($\mathcal{A}_{\text{focus}}$) of models after SFT and FIT on the SS dataset. Here, $C$ refers to the causal feature (sentiment) and $S$ the spurious feature (presence of one of the keywords of "Bayesian" and "Pineapple").

classes, and exclude examples with neutral labels from our augmented dataset. The feature set is given as $\mathcal{F} = \{$*sentiment, presence of keywords ["Bayesian", "Pineapple"]*$\}$. We inject these features so that the presence of "Pineapple" and "Bayesian" are spuriously correlated with negative and positive sentiment, respectively. The degree of co-occurrence is governed by $\rho_{\text{spurious}}$, which varies according to the test sets described in Section 3. We ensure that $\rho_{\text{spurious}}$ is the same for all feature values within each dataset split. In particular, we set $\rho_{\text{spurious}}$ to be $0.5, 0.5, 0.9, 0.25$ and $0.9$ on $\mathcal{D}_{\text{train}}, \mathcal{D}_{\text{iid}}, \mathcal{D}_{\text{high}}, \mathcal{D}_{\text{low}}$ and $\mathcal{D}_{\text{flipped}}$ respectively. Further details of the SS dataset can be found in Appendix A.7.

**Results.** Figure 3 shows the focus accuracy results of the three LLMs on the SS dataset after SFT and after FIT. We see that across all focus instructions and all models, FIT shows significant improvement over the baselines, achieving very high focus accuracy.

> ***Key takeaways.*** High focus accuracy on SS indicates that FIT training successfully allows the model to alter its response by considering the feature on which it is instructed to focus or not focus. This shows that the model's behaviour can be effectively steered using FIT.

### 4.2 FIT PERFORMS WELL WITH MORE COMPLEX FEATURES ON THE SMNLI DATASET AND GENERALISES UNDER DISTRIBUTION SHIFT

**Spurious MNLI dataset (SMNLI).** Next, we evaluate our method on a more complex dataset with subtler features. Specifically, we construct an NLI dataset by sub-sampling from MNLI (Williams et al., 2018), where we induce a spurious correlation between text genres and labels by sub-sampling accordingly. We refer to this dataset as SMNLI, where the feature set is defined as $\mathcal{F} = \{$*NLI relationship, genre*$\}$. The co-occurrence rate of genres and their spuriously associated label is governed by $\rho_{\text{spurious}}$, which varies across the test sets discussed in Section 3. We ensure that $\rho_{\text{spurious}}$ is the same for all feature values within each dataset split. In particular, we set $\rho_{\text{spurious}}$ to be $1/3, 1/3, 0.9, 0.1$ and $0.9$ on $\mathcal{D}_{\text{train}}, \mathcal{D}_{\text{iid}}, \mathcal{D}_{\text{high}}, \mathcal{D}_{\text{low}}$ and $\mathcal{D}_{\text{flipped}}$ respectively.

Moreover, for SMNLI, we hold out specific genres at test time to evaluate our model's ability to generalise under distribution shift when feature values change. We do this by sub-sampling a held-out portion of the MNLI dataset. During training, we use three selected genres (government, fiction, and travel) to evaluate our models. At test time, we add an additional three held-out genres (faceto-face, nineeleven, and verbatim). We again ensure that $\rho_{\text{spurious}}$ is constant within each dataset split

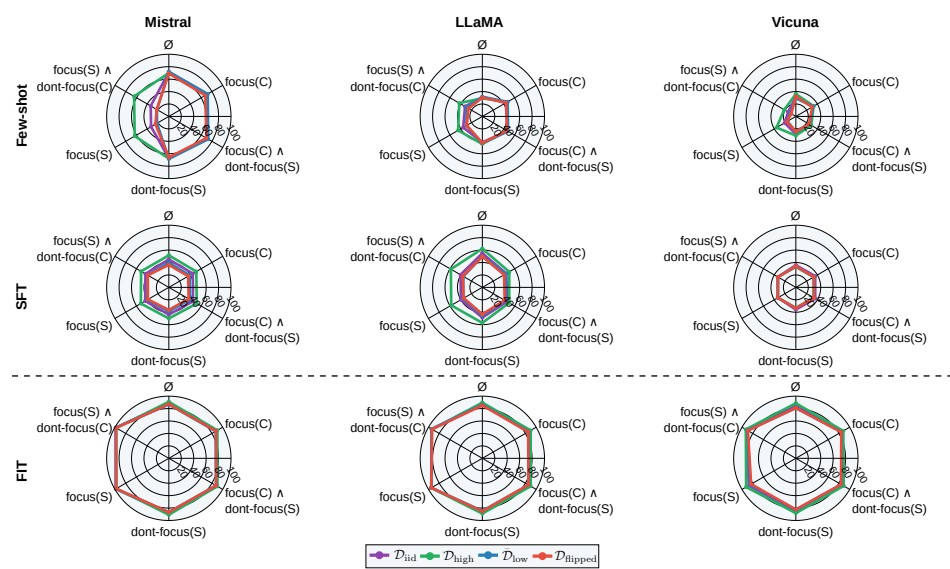

Figure 4: **SMNLI focus accuracies (↑).** Focus accuracy ($\mathcal{A}_{\text{focus}}$) of models after SFT and FIT on the SMNLI dataset. Here, $C$ refers to the causal feature (logical relationship between premise and hypothesis) and $S$ the spurious feature (genre of the underlying text)

for all feature values, and use the same set of corresponding $\rho_{\text{spurious}}$ as within the SMNLI test sets described above. Further details of the SMNLI dataset can be found in Appendix A.9. **Results.** Figure 4 depicts the focus accuracy results of the three models on the SMNLI test splits. We observe that even for the more complex feature of genre, FIT achieves very high focus accuracy, significantly improving over the baselines. This demonstrates that FIT effectively trains the model to handle more complex features, allowing it to dynamically focus on or disregard these features when making predictions.

Figure 5 shows the focus accuracy of models on the distribution-shifted test sets across different values of $\rho_{\text{spurious}}$. When focusing on the causal feature or ignoring the spurious feature, the model maintains strong performance in terms of focus accuracy, even on unseen genre values (over 80% focus accuracy for FIT models on the second row of Figure 5). Note that, while we observe low focus accuracy when focusing on spurious features, this is expected, as the spurious labels associated with these new genres were not encountered during training. Thus when focusing on these features the model does not know what label to predict. This result highlights that the focus-tuned models have indeed learned spurious associations during training and correctly reproduces them when instructed to focus on these spurious features, even for new spurious feature values. When instructed to focus on the causal feature (or even just to ignore the spurious feature), the model still shows strong generalisation in the presence of distribution shift.

> ***Key takeaways.*** FIT achieves high focus accuracy on more complex features and maintains strong performance under distribution shift in terms of feature values. This demonstrates FIT's ability to generalise to new contexts and reliably handle changing feature values, which is crucial for ensuring consistent and robust model performance in dynamic settings.

### 4.3 FIT STEERS BEHAVIOUR IN THE PRESENCE OF SOCIAL BIAS DATA AND GENERALISES TO UNSEEN FEATURES

**Bias Benchmark for QA (BBQ) dataset**. Finally, we experiment with BBQ Parrish et al. (2022), a widely-utilised multiple-choice question-answering benchmark annotated with nine forms of social bias that are relevant to any given answer, such as stereotypes that would imply a given answer to an otherwise ambiguous question (see Figure 1). The feature set contains $\mathcal{F} = \{$*question context, gender identity, race/ethnicity, ..., disability status*$\}$, which contains one causal feature (question context) and 9 bias features. Of the $n = 9$ bias features, we focus-tune mod-

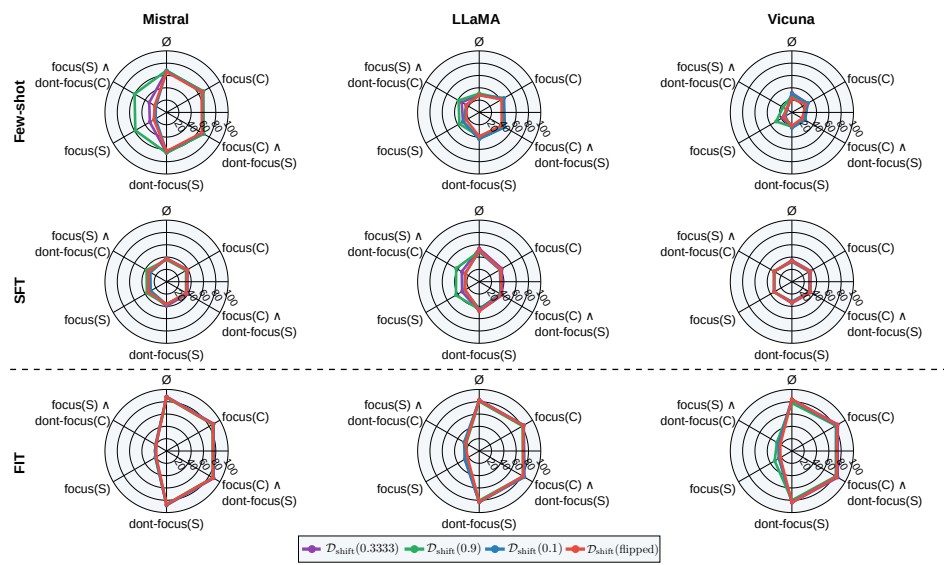

Figure 5: **SMNLI focus accuracies (↑) under distribution shift .** Focus accuracy ($\mathcal{A}_{\text{focus}}$) of models after SFT and FIT on SMNLI, evaluated on distribution-shifted test sets with different feature values (genres) from training. $\text{shift}(\rho_{\text{spurious}})$ refers to test sets where feature values (genres) differ from training with a co-occurrence rate of $\rho_{\text{spurious}}$. Here, "flipped" indicates a change in spurious label associations between training and testing, with high $\rho_{\text{spurious}}$ in the test set.

els with respect to 6, and test on these 6 features plus the remaining 3 bias features in order to test how well FIT generalises to features that are not seen during focus tuning. Here, we consider the spurious features to be the presence of a particular social group (e.g., men or women) in the question context, and spurious answers to be those that would be indicated by relying on social stereotypes rather than the specific question context (e.g., see Figure 1). The stereotyped response used to determine spurious answers for these bias features are provided as part of the BBQ dataset.

**Results.** Figure 6 shows the focus accuracy results of the three models on the BBQ dataset, visualising performance on features seen during training and unseen, held-out features. The models demonstrate high and comparable focus accuracy across both seen and unseen bias features, indicating that FIT generalises well to unseen features, including nuanced reasoning about group stereotypes. This highlights the usefulness of FIT in mitigating social biases in LLM responses. Specifically, FIT can effectively learn, reason about, and rule out biases when formulating responses, making it a practical tool for bias mitigation.

> *Key takeaways.* FIT can effectively teach models to adjust their responses based on knowledge of social biases. This ability generalises to biases not seen during FIT training, indicating FIT's utility for bias mitigation.

## 5 ABLATION

**Generalisation to different test-time prompt formats.** As observed in the IT literature, instruction-tuned models sometimes memorise instruction formats and struggle to follow paraphrased instructions at test time (Ghosh et al., 2024). In Appendix A.6 (Figure 8), we compare the performance of models on the SMNLI dataset when using the same focus instructions at training and test time versus using paraphrased instructions at test time. We generate 10 different test-time focus instructions of each instruction type defined in Equation (1) by paraphrasing the existing focus instruction using ChatGPT (OpenAI, 2022). The results show minimal variation in focus accuracy across different dataset splits and focus features, even when testing on paraphrased prompts, indicating that FIT indeed teaches models a general capacity to focus on or ignore features regardless of the specific way that focus instructions are phrased.

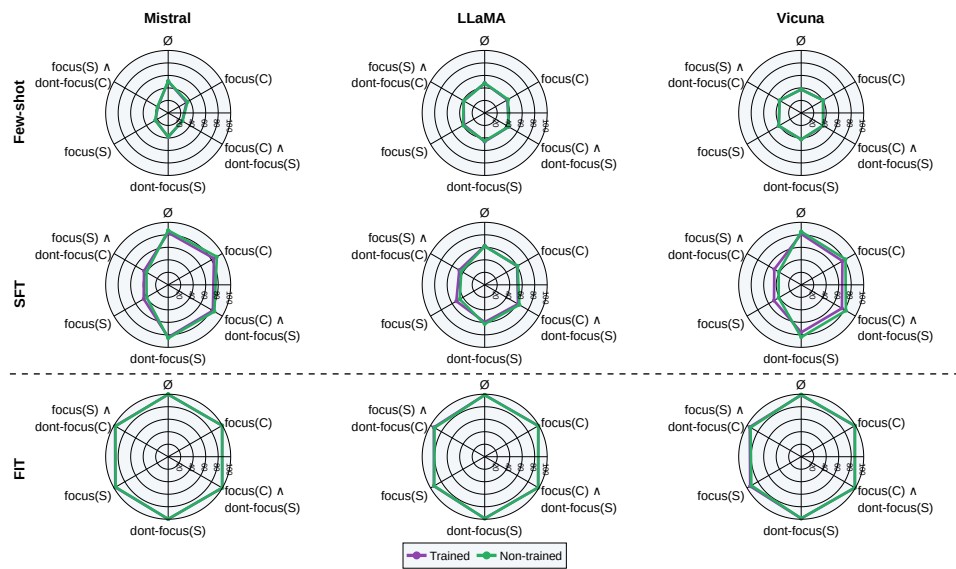

Figure 6: **BBQ focus accuracies** ($\uparrow$). Focus accuracy ($\mathcal{A}_{\text{focus}}$) of models after SFT and FIT on the BBQ dataset. We include focus accuracy evaluated on bias features seen at training time (purple) and on held-out bias features seen only at test time (green).

## 6  CONCLUSIONS

In this work, we introduce Focus Instruction Tuning (FIT), a method designed to steer the behaviour of LLMs by focusing on or ignoring specific features when formulating responses. Across a range of tasks and settings, we demonstrate that FIT can be used to steer LLM behaviours at inference time, even in the context of distribution shifts over feature values or when generalizing to unseen features at inference time. Additionally, we show that our method can mitigate biases by identifying and factoring out known stereotypes that might otherwise influence responses. Thus, FIT represents a step toward enabling more robust, fair, and controllable LLMs.

We recommend that future work explore the effectiveness of FIT across a broader variety of tasks, including open-ended, free-form natural language generation tasks such as summarization or translation. Another promising direction is investigating whether FIT can generalise not only across features but also across different categories of tasks (cf. FLAN; Longpre et al. 2023).

## ETHICAL CONSIDERATIONS AND LIMITATIONS

The ability to dynamically steer model behaviour by focusing on or ignoring features, as enabled by FIT, holds significant potential for reducing algorithmic discrimination and mitigating harms. Practitioners can leverage FIT to identify and correct biases by measuring discrepancies in behaviour when a model focuses on or ignores specific features. Additionally, FIT enhances explainability by attributing model predictions to input features, enabling more transparent and productive human-AI collaboration. This supports ethical and responsible decision-making by assessing whether predictions are justified. FIT also enhances robustness by prioritising stable causal features expected to generalise across domains while ignoring spurious, domain-specific biases, making it a valuable tool for fairness, explainability, and robustness.

However, risks include potential misuse by bad actors to bias models, though this is not unique to FIT and could already be achieved through biased fine-tuning. Additionally, as noted in Appendix A.10, FIT may face challenges when addressing features that overlap heavily or lack distinctiveness. While these constraints may arise in specific contexts, they do not diminish FIT's broader applicability across natural-language tasks.

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

## A    APPENDIX

### A.1    FT TRAINING AND OPTIMISATION SETTINGS

**FT Optimisation.** Algorithm 1 gives precise details on how we implement FIT in practice when performing SFT of a model on a given training set. In particular, it shows how we approach optimising the FIT training objective given in Equation (3).

---

**Algorithm 1** Algorithm for Focus Instruction Tuning (FIT) Training Procedure to Optimise Equation (3).

---

1: **Input:** Dataset $\mathcal{D} = \{(x_i, y_i)\}_{i=1}^N$, The feature set contains $\mathcal{F}$, instruction $I$, model parameters $\theta$, batch size $B$, number of epochs $E$, step size $\eta$, and mapping function $y_{\text{focus}} = y_{\text{focus}}(x, y, I_{\text{focus}})$.
2: **Initialise:** Model parameters $\theta$, optimiser
3: **for** epoch = 1 to $E$ **do**
4:   **for** mini-batch $\{(x^b, y^b)\}_{b=1}^B$ from $\mathcal{D}$ **do**
5:     **for** each $(x^b, y^b)$ in the mini-batch **do**
6:       Sample focus instruction $I_{\text{focus}}^b \sim p_{\mathcal{I}_{\text{focus}}}(\mathcal{I}_{\text{focus}})$
7:       Compute $y_{\text{focus}}^b = y_{\text{focus}}(x^b, y^b, I_{\text{focus}}^b)$
8:     **end for**
9:     Compute average loss given through empirical estimator of the loss defined in Equation (3) over the batch:

$$L(\theta) = \frac{1}{B} \sum_{b=1}^B - \log p_\theta(y_{\text{focus}}^b | I, I_{\text{focus}}^b, x^b)$$

10:     Update model parameters $\theta$ using optimiser:

$$\theta \leftarrow \theta - \eta \nabla_\theta L(\theta)$$

11:   **end for**
12: **end for**
13: **Output:** Optimised model parameters $\theta$

---

**FT training settings.** We use the SFTTrainer class from HuggingFace (Wolf et al., 2020) and use all of the default training settings for performing SFT of LLMs. Furthermore, we define $p(\mathcal{I}_{\text{focus}})$ by placing a small probability (in our experiments, 0.05) on the empty focus instruction $\emptyset$. We then uniformly distribute the remaining probability mass over the non-empty focus instructions.

We implement early stopping on a held-out validation set based on the cross-entropy loss over focus labels $y_{\text{focus}}$ corresponding to randomly sampled focus instructions - this matches the context in which the models will be evaluated. We obtain this set by splitting our training set in an 80/20% for training and validation. We use a patience of 4 validation evaluation steps, which occur after a fixed number of steps.

We use LoRA (Hu et al., 2021) for parameter-efficient fine-tuning. We target the query and value projection matrices within each LLM and use LoRA $r = 16$ and $\alpha = 32$ across models.

**Choice of $\rho_{\text{spurious}}$ during training.** Consider a classification problem with $N$ classes so that $|\mathcal{Y}| = 6$. During FIT training we want the model to learn to change it's behaviour depending on which features are specified to be focused on or ignored.

To achieve this, consider when the focus instruction requires that the model focus on a spurious feature during training, which means that $I_{\text{focus}} \in \{\text{focus}(S), \text{focus}(S) \land \text{ignore}(F_j) \mid F_j \in \mathcal{F} \setminus \{S\}\}$, we choose $\rho_{\text{spurious}} = 1/N$. The definition of $\rho_{\text{spurious}}$ given in Equation (4) implies that in this setting $\rho_{\text{spurious}} = \mathbb{P}(Y = y_s | X, I, I_{\text{focus}}, s \in X)$ for a feature value $s$ of spurious feature $S$ in input $X$. Therefore, setting $\rho_{\text{spurious}} = 1/N$ for the training dataset induces a uniform distribution over the set the set of class labels conditioned on an input $X$ and focus instruction $I_{\text{focus}}$ during training.

The entropy of this discrete distribution is then given by:

$$\mathcal{H}(\mathbb{P}(Y = y_s | X, I, I_{\text{focus}}, s \in X)) = \tag{6}$$

$$= \mathbb{E}_{y \sim \mathbb{P}(Y = y_s | X, I, I_{\text{focus}}, s \in X)} \left[ -\log \mathbb{P}(Y = y_s | X, I, I_{\text{focus}}, s \in X) \right] \tag{7}$$

$$= -\sum_{y \in \mathcal{Y}} \mathbb{P}(Y = y_s | X, I, I_{\text{focus}}, s \in X) \log \mathbb{P}(Y = y_s | X, I, I_{\text{focus}}, s \in X) \tag{8}$$

$$= -\sum_{y \in \mathcal{Y}} \log \left( \frac{1}{N} \right) \frac{1}{N} \tag{9}$$

$$= -\log \left( \frac{1}{N} \right) \tag{10}$$

$$= \log N. \tag{11}$$

It is well known that discrete uniform distributions have maximum entropy (Bishop & Nasrabadi, 2006).

Practically, this means that the causal feature is not predictive of the focus label, which is what we are training the model to predict. A lower-entropy distribution (compared to the uniform distribution) during training would result in a higher co-occurrence of the spurious labels with the underlying causal labels. This could lead the model to rely on the underlying task-causal feature to solve the task, rat her than learning to adapt its behaviour when focusing on a non-causal feature. Therefore, using the maximum-entropy uniform distribution during training better enables the model to learn to adjust its behaviour based on the specified features. This ensures that the model does not fall back on causal features when spurious ones are the focus, thus improving the steerability of the model.

## A.2 EVALUATION METRICS

**Generation settings.** We generate responses from our FT model using constrained beam-decoding (Anderson et al., 2017) with 8 beams. This ensures that the answer labels for each classification task that we investigate appear in the model's output. We limit the maximum number of newly generated tokens to be 5 to stop any unnecessary text given after the model's initial classification prediction.

**Computing the focus accuracy metric.** We report the accuracy of generations when evaluating FT models. As we are guaranteed to include the task labels within the model's response through constrained decoding, we simply check to see if the focus label, $y_{\text{focus}}$, is within the model's response or not in order to determine if the model's response is correct.

## A.3 FIT FOCUS INSTRUCTIONS AND PROMPT TEMPLATES

**Focus instructions.** We consider the following focus instruction formats for the different focus instructions introduced in Equation (1) which are used for FIT training and evaluation:

---

**Focus instructions** $\mathcal{I}_{\text{focus}}$

For features $F_i, F_j \in \mathcal{F}$:

**Focus instructions** $\text{focus}(F_i)$:

- Direct your attention solely to $F_i$.
- Concentrate all your reasoning on $F_i$.
- Make $F_i$ the central factor in your decision.
- Base your judgment exclusively on $F_i$.
- Pay attention only to $F_i$ when making your prediction.
- Use $F_i$ as the key input for your evaluation.
- Focus entirely on $F_i$ and ignore other aspects.
- Rely exclusively on $F_i$ to reach your conclusion.
- Consider only $F_i$ and disregard all else.
- Let $F_i$ be the primary basis for your decision.

**Ignore instructions** $\text{ignore}(F_i)$:

- Completely rule out $F_i$ from your reasoning.
- Disregard any influence of $F_i$ in your prediction.
- Treat $F_i$ as irrelevant to your decision-making process.
- Exclude $F_i$ entirely from your evaluation.
- Do not let $F_i$ play any role in your assessment.
- Intervene to prevent $F_i$ from affecting your prediction.
- Ensure that $F_i$ has no bearing on your final decision.
- Block $F_i$ from contributing to your reasoning.
- Negate the impact of $F_i$ in your prediction.
- Ruling out $F_i$ is crucial—do not let it affect your decision.

**Focus and Ignore instructions** $\text{focus}(F_i) \wedge \text{ignore}(F_j)$

- Focus specifically on $F_i$. Disregard $F_j$ in your decision-making process.
- Base your prediction solely on $F_i$. Exclude $F_j$.
- Direct all your attention to $F_i$. Block out $F_j$ from your prediction.
- Consider only $F_i$ in your reasoning. Rule out $F_j$ in your decision-making.
- Prioritize $F_i$. Completely ignore $F_j$ in your prediction.
- Do not consider $F_j$ in your decision-making process. Focus exclusively on $F_i$.
- Ignore any influence of $F_j$. Concentrate on $F_i$ in your prediction.
- Disregard $F_j$ entirely. Base your analysis solely on $F_i$.
- Rule out $F_j$ in your prediction. Shift your focus to $F_i$.
- Do not pay attention to $F_j$ in your decision-making process. Rely only on $F_i$.

**Prompt template for SS.** We consider the following prompt templates for the SS dataset:

---

**SS Focus instruction prompt templates** $\mathcal{I}_{\text{focus}}$

```
<INSTRUCTION>
You are a language model performing sentiment analysis on a binary dataset, making predictions from the labels [negative,
positive]. Make your prediction based on the relevant features described below, focusing on the specified features and ignoring
those deemed irrelevant. For the input below, output either negative or positive ONLY for your prediction of the input's label.
<END OF INSTRUCTION>

<FEATURE CONSIDERATIONS>
{focus instruction - Ifocus}
<END FEATURE CONSIDERATIONS>

### Input:  {input}

### Output:
```

where for the instruction with no focus instruction, $\emptyset$, we simply drop the feature considerations section and focus instruction from the prompt.

**Prompt template for SMNLI.** We consider the following prompt templates for the SMNLI dataset.

---

**NLI focus instruction prompt template $\mathcal{I}_{\text{focus}}$**

```
<INSTRUCTION>
You are a language model performing natural language inference (NLI) on a dataset with three labels [entailment, neutral,
contradiction]. For the inputs consisting of premise and hypothesis below, make your prediction based on the relevant features
described, focusing on the specified features and ignoring those deemed irrelevant. Output either entailment, neutral, or
contradiction ONLY for your prediction of the input's label.
<END OF INSTRUCTION>

<FEATURE CONSIDERATIONS>
{focus instruction - I_focus}
<END FEATURE CONSIDERATIONS>

### Premise:  {premise}
### Hypothesis:  {hypothesis}

### Output:
```

---

where we again drop the feature considerations section and focus instruction from the prompt to form the instruction prompt $\emptyset$ with no focus instruction.

**BBQ focus instruction prompt template $\mathcal{I}_{\text{focus}}$** We consider the following prompt templates for the BBQ dataset.

---

**BBQ focus instruction prompt template $\mathcal{I}_{\text{focus}}$**

```
<INSTRUCTION>
You are a language model answering a QUESTION based on a provided CONTEXT, choosing to ANSWER with one of three
options: 1, 2, or 3. Each option corresponds to an answer from the provided ANSWER OPTIONS. You must output EXACTLY
one of these numbers (1, 2, or 3) representing the answer you select.
<END OF INSTRUCTION>

<FEATURE CONSIDERATIONS>
{focus instruction - I_focus}
<END FEATURE CONSIDERATIONS>

### CONTEXT: {context}

### QUESTION: {question}

### ANSWER OPTIONS:
(1) {ans0}
(2) {ans1}
(3) {ans2}

### ANSWER:
```

---

where we again drop the feature considerations to get the template for the focus instruction $\emptyset$.

## A.4  BASELINES

We include results for the following two baselines to fruther supplement the results presented in the main section of the paper.

**SFT($y_{\text{focus}}$) baseline.** We implement an SFT baseline that follows the same training procedure as FIT, except during training, we exclude any focus instructions from the input prompts while still training on the focus labels. This provides a fair comparison with FIT, as the models are trained on the same input text and label pairs. The rest of the training setup, including hyperparameters and early stopping, remains identical to the FIT training setup.The model is tested on the full set of focus instructions prompts detailed in Equation (1).

**Few-shot baseline.** This second baseline compares FIT training to few-shot inference using the original pre-trained models without additional fine-tuning on our spurious datasets. Specifically, we use 5 in-context examples across all datasets. For the in-context examples, we concatenate multiple examples one after the other, including the instructional prompt only for the first in-context example and the final test example. Each in-context example contains the same focus instruction as the test example for which they serve as context. The model is tested on the full set of focus instructions prompts detailed in Equation (1).

For completeness, we report two additional baselines: vanilla SFT and zero-shot baselines.

**SFT($y$) baseline.** We implement a vanilla SFT baseline that simply trains a model using SFT on inputs and their ground truth labels (as opposed to focus labels in the SFT($y_{\text{focus}}$) baseline). During training, only standard IT prompts are used, with no additional focus instructions included. The rest of the training setup, including hyperparameters and early stopping, remains identical to the FIT training setup. The model is tested on the full set of focus instructions prompts detailed in Equation (1).

**Zero-shot baseline.** Finally, we include a zero-shot inference baseline using the original pre-trained models without additional fine-tuning on our spurious datasets. No in-context examples are used at inference time, and the model is not trained at all beyond it's pre-training. The model is tested on the full set of focus instructions prompts detailed in Equation (1).

### A.5 Additional Baselines Results

In this section, we include the two additional baselines -SFT($y$) and zero-shot - on the SMNLI dataset to further supplement the results in Section 4.2.

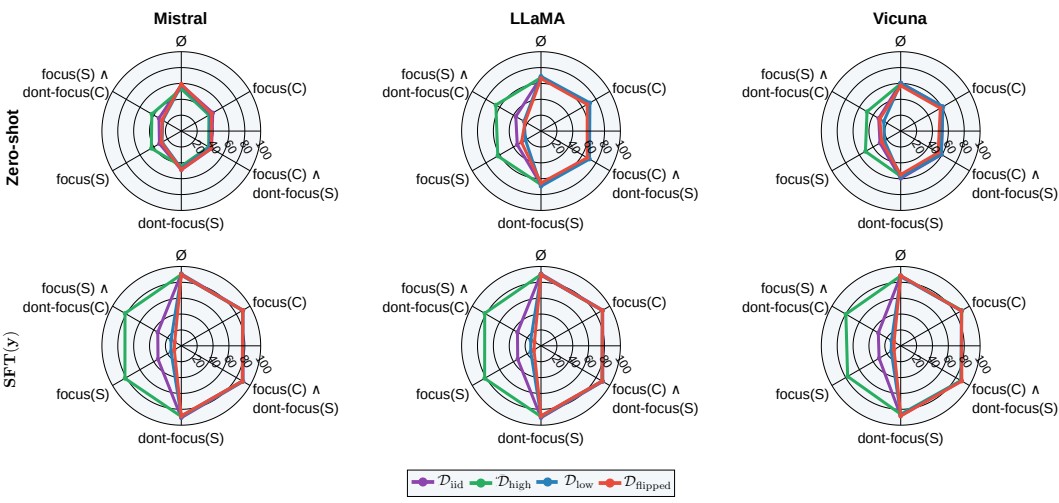

Figure 7: **Focus accuracy ($\uparrow$) for zero-shot and SFT($y$) on SMNLI.** Figure giving focus accuracies ($\mathcal{A}_{\text{focus}}$) of the additional zero-shot and SFT($y$) baselines on the SMNLI dataset.

### A.6 SMNLI Ablation of Training and Test time Focus Instruction Rephrasing Differences

We analyse the impact of using the same versus different sets of focus instructions at training and test time when applying FIT models. Specifically, we generate alternative test set focus instructions by paraphrasing the training focus instructions, as shown in Appendix A.3, using ChatGPT.

As depicted in Figure 8, the results of this ablation reveal negligible differences between using the same or different focus instruction phrasings during training and testing. This indicates that FIT effectively trains the model to focus on or ignore features, regardless of how the instructions are phrased.

### A.7 Spurious Sentiment (SS) Dataset

We take a pre-existing dataset, in this case SST-5 (Socher et al., 2013a), and modify it in order to induce a known spurious feature and create a spurious binary sentiment analysis dataset.

**Data-generating process (DGP).** We frame our DGP using a graphical model to describe the synthetic dataset that we create. We follow a similar model to that described in (Arjovsky et al., 2019),

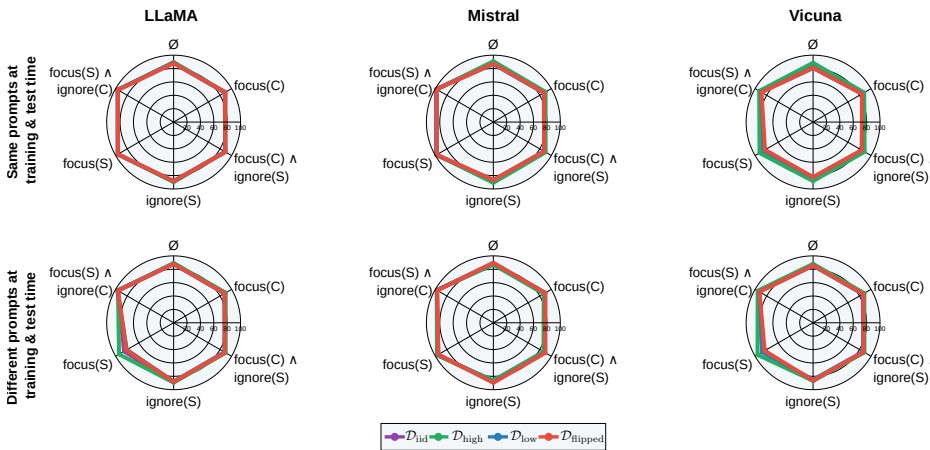

Figure 8: **Focus accuracy** (↑) **for different training and test focus instruction sets.** Figure comparing focus accuracies ($\mathcal{A}_{\text{focus}}$) of sampling from the same (left) and different (right) sets of focus instructions at training and test time of models on the SMNLI dataset.

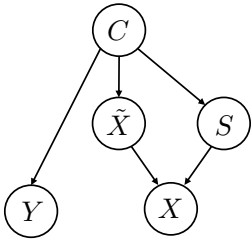

Figure 9: **SS DGP**. Graphical model showing the data generating process for modifying examples from the SST-5 dataset to introduce a new spurious keyword feature $S$.

specifically the model used for generating their coloured MNIST dataset. We use the following variables within our graphical model:

- $C$ - true underlying sentiment, the causal feature within this task, sampled from the original dataset.

- $S$ - spurious feature, here this is the presence of the keywords *Bayesian* or *Pineapple*. We represent this as a binary vector $S \in \{0,1\}^2$, where the first and second components of this vector denote the presence of either the keyword *Pineapple* or *Bayesian* respectively.

- $\tilde{X}$ - is a sampled example from the original dataset that we are modifying to inject known spurious correlations.

- $X$ - original example $\tilde{X}$ but augmented to include the spurious feature.

- $Y$ - final label for element $X$.

The graphical model describing the DGP of the SS dataset is given in Figure 9. This admits a functional representation in the form:

$$C = f_C(U_C); \tag{12}$$
$$\tilde{X} = f(C, U_{\tilde{X}}); \tag{13}$$
$$S = f_S(C, U_S); \tag{14}$$
$$X = f_X(\tilde{X}, S, U_X, U_{\text{incld.}}); \tag{15}$$
$$Y = f_Y(C, U_Y). \tag{16}$$

where $U_{(\cdot)}$ are variables introducing sources of randomness into the generating process. More explicitly, we consider the following set of equations, where $\mathcal{D}$ denotes the underlying dataset that we are manipulating:

$$C \sim \texttt{Ber}(\rho_C), \text{ where } \rho_C = \rho_C(\mathcal{D}); \tag{17}$$

$$\tilde{X} \sim p_{\mathcal{D}}(\cdot|C), \tag{18}$$

$$S = (\mathbf{1}_{C=0}, \mathbf{1}_{C=1}); \tag{19}$$

$$U_{\text{incld.}} \sim \texttt{Ber}(\rho_{\text{incld.}}); \tag{20}$$

$$U_X \sim \texttt{Ber}(\rho_{\text{spurious}}); \tag{21}$$

$$X = \begin{cases} U_X \texttt{LLM}(\tilde{X}, S) + (1 - U_X)\texttt{LLM}(\tilde{X}, 1 - S) & \text{if } U_{\text{incd.}} = 1 \\ \tilde{X} & \text{if } U_{\text{incd.}} = 0 \end{cases}; \tag{22}$$

$$Y = C, \tag{23}$$

The variable $\rho_C$ gives the distribution of sentiment labels in the original binarised SST-5 dataset. Moreover, $p_{\mathcal{D}}(\cdot|C)$ denotes the conditional dataset distribution of the different input texts give $C$ (here we assume that we are just uniformly sampling text with the given sentiment $C$) and $\mathbf{1}_{(\cdot)}$ denotes the indicator function. In addition, $\rho_{\text{incl.}}$ gives the proportion of spurious features that are included within original dataset examples. This corresponds to proportion of examples within the original dataset that are modified by the above process and therefore contain spurious feature values.

Finally, we prove that $\rho_{\text{spurious}}$ gives the concurrence rate between the label $Y$ and the spurious feature values of $S$. The proof rests on the fact that $U_X$, which gives whether a spurious feature value $s$ is injected into $\tilde{X}$ or whether the other value $1-s$ is injected instead, controls the cooccurrence between $Y$ and the spurious feature value $s$. In particular, we note that $\mathbb{P}(Y = y, U_X = 1, X \neq \tilde{X}|\tilde{X}, S = s)$ gives the co-occurrence rate between the label $y$ and spurious feature $s$ in the dataset (assuming that if the feature is only present within the dataset through the inclusion of the feature and not within the original dataset examples).

**Proposition 1.** *From the DGP described in Figure 9, we have that*

$$\mathbb{P}(Y = 1, U_X = 1, X \neq \tilde{X}|\tilde{X}, S = (0,1)) = \rho_{incdl.} \cdot \rho_{spurious}; \tag{24}$$

$$\mathbb{P}(Y = 0, U_X = 1, X \neq \tilde{X}|\tilde{X}, S = (1,0)) = \rho_{incdl.} \cdot \rho_{spurious}. \tag{25}$$

*Proof.* Using the chain rule of probability, we see that

$$\mathbb{P}(Y = 1, U_X = 1, X \neq \tilde{X}|\tilde{X}, S = (0,1)) = \tag{26}$$

$$= \mathbb{P}(Y = 1|U_X = 1, \tilde{X}, X \neq \tilde{X}, S = (0,1))\mathbb{P}(U_X = 1, X \neq \tilde{X}|\tilde{X}, S = (0,1)) \tag{27}$$

$$= \mathbb{P}(Y = 1|S = (0,1))\mathbb{P}(U_X = 1, X \neq \tilde{X}|\tilde{X}, S = (0,1)) \tag{28}$$

$$= \mathbb{P}(Y = 1|S = (0,1))\mathbb{P}(U_X = 1|\tilde{X}, X \neq \tilde{X}, S = (0,1))\mathbb{P}(X \neq \tilde{X}|\tilde{X}, S = (0,1)) \tag{29}$$

$$= \mathbb{P}(Y = 1|S = (0,1))\mathbb{P}(U_X = 1)\mathbb{P}(X \neq \tilde{X}|\tilde{X}) \tag{30}$$

$$= \mathbb{P}(Y = 1|C = 1)\mathbb{P}(U_X = 1)\mathbb{P}(U_{\text{incld.}} = 1) \tag{31}$$

$$= 1 \cdot \rho_{\text{incdl.}} \cdot \rho_{\text{spurious}} \tag{32}$$

$$= \rho_{\text{incdl.}} \cdot \rho_{\text{spurious}}, \tag{33}$$

where we have used that $S$ and $C$ share a deterministic relationship and have used the conditional independencies specified within the graphical model depicted in Figure 9, and through the noise terms $U_{(\cdot)}$ in the SCEs introduced above. $\square$

With $\rho_{\text{incld.}} = 1$, we have that, indeed, the co-occurrence rate between the presence of spurious feature values and the labels $Y$ is given by $\rho_{\text{spurious}}$.

We now connect this to predicitvity which is defined Equation (4) in order to justify the notation that we are using within the SCEs above.

**Corollary 1.** *From the DGP described in Figure 11, we have that*

$$\mathbb{P}(Y = 1|\tilde{X}, S = (0,1), U_X = 1, X \neq \tilde{X}) = \rho_{spurious}; \tag{34}$$

$$\mathbb{P}(Y = 0|\tilde{X}, S = (1,0), U_X = 1, X \neq \tilde{X}) = \rho_{spurious}. \tag{35}$$

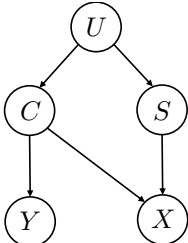

Figure 10: **SS SCM.** SCM showing showing the spurious correlation present between the keyword feature $S$ and the label $Y$ of examples within the SS dataset, induced through the described data augmentation process.

*These events correspond to the events given in Equation (4) that define $\rho_{spurious}(s)$ generally. Therefore, the notation of $\rho_{spurious}$ in Equation (4) is justified.*

*Proof.* Using Proposition 1 and the definition of conditional probability, we get that

$$\mathbb{P}(Y = 1 \mid \tilde{X}, S = (0,1), U_X = 1, X \neq \tilde{X}) \tag{36}$$

$$= \frac{\mathbb{P}(Y = 1, U_X = 1, X \neq \tilde{X} \mid \tilde{X}, S = (0,1))}{\mathbb{P}(U_X = 1, X \neq \tilde{X} \mid \tilde{X}, S = (0,1))} \tag{37}$$

$$= \frac{\mathbb{P}(Y = 1, U_X = 1, X \neq \tilde{X} \mid \tilde{X}, S = (0,1))}{\mathbb{P}(U_X = 1 \mid \tilde{X}, S = (0,1), X \neq \tilde{X})\mathbb{P}(X \neq \tilde{X} \mid \tilde{X}, S = (0,1))} \tag{38}$$

$$= \frac{\mathbb{P}(Y = 1, U_X = 1, X \neq \tilde{X} \mid \tilde{X}, S = (0,1))}{\mathbb{P}(U_X = 1)\mathbb{P}(X \neq \tilde{X} \mid \tilde{X}, U_X = 1)} \tag{39}$$

$$= \frac{\rho_{\text{icld.}} \cdot \rho_{\text{spurious}}}{\rho_{\text{icld.}} \cdot 1} \tag{40}$$

$$= \rho_{\text{spurious}}, \tag{41}$$

where we have used the independencies specified within the DGP in Figure 11 and the noise term $U_x$ within the functional description of this DGP. $\square$

**SCM from this DGP.** Through the above data generation process, we introduce a new spurious feature within the dataset $S$, the presence of the keywords *Bayesian* and *Pineapple*. Recalling that $S = (1,0)$ and $S = (0,1)$ correspond to insertion of the keywords *Pineapple* and *Bayesian* respectively, we introduce the following spurious correlations between feature values of $S$ and label $Y$:

1. The presence of the word *Pineapple* in the text $X$ is spuriously correlated the label 0 (negative sentiment).

2. The presence of the word *Bayesian* in the text $X$ is spuriously correlated with the label 1 (positive sentiment).

The sentiment feature still remains causal within the augmented SS dataset, fully predicting the label $Y$ for each dataset example.

The above DGP, through the introduction of spurious feature $S$, induces a SCM that describes the spurious correlation between spurious feature $S$ and the label $Y$. The SCM, shown in Figure 10, follows the style-content decomposition described in (Kaddour et al., 2022), where $U$ is some hidden confounding variable.

**Data generation methodology.** We use `Llama-3.1-70B-Instruct` to generate modifications $X$ of original dataset examples $\tilde{X}$ to create new text which include the new keywords feature. The prompt we use for generation when modifying examples to include spurious features is give as:

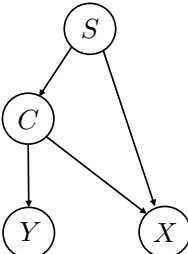

Figure 11: **SMNLI DGP**. Graphical model showing the data generating process for modifying examples from the MNLI dataset to introduce a new spurious keyword feature $S$.

---

**Data augmentation prompt**

You are a language model designed to modify a piece of text to include an additional feature in a simple, natural way while keeping your output as similar as possible to the original text.

**Features**
- pineapple: Include the word 'pineapple'.
- Bayesian: Include the word 'Bayesian'.

**Instructions**
- Ensure the output is grammatically correct.
- Keep the output as similar as possible to the original text.
- Make the minimal number of modifications and add the fewest new tokens possible to satisfy the chosen feature.
- Do not change the sentiment of the original text.
- Do not significantly alter the length of the output.
- Incorporate the feature naturally within the original text so that it blends seamlessly with the text's context.
- Do not only append additional clauses at the end of the text to include the feature.
- Inclusions should be case sensitive, e.g., include 'Bayesian' BUT NOT 'bayesian'.

**Output**
- Only return the modified text, with no additional explanations or reasoning.
- Should strictly follow the feature description and the set of instructions.
- Only include the one feature given; the other features SHOULD NOT be included even accidentally.

---

A.8 SPURIOUS NLI DATASET (SMNLI)

We generate a tertiary NLI dataset, SMNLI, with a known spurious feature. We do this considering the MNLI dataset Williams et al. (2018). This is a NLI dataset with three labels: entailment (0), neutral (1) and contradiction (2), where data is sampled from 5 underlying categories or genres (telephone, government, travel, fiction or slate). We aim to induce spurious correlations between the underlying genres and labels.

**Data-generating process (DGP).** We consider a graphical model to describe the DGP of examples within the SMNLI dataset. We use the following variables within our DGP:

- $C$ - NLI relationship between a premise and hypothesis pair, the causal feature within this task, sampled from the original dataset.

- $S$ - spurious feature, here this is the genre of the premise and hypothesis. This is a categorical variable.

- $X$ - example from the MNLI dataset.

- $Y$ - final label for element $X$.

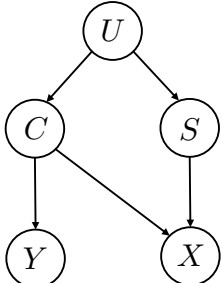

Figure 12: **SMNLI SCM.** SCM showing showing the spurious correlation present between the keyword feature $S$ and the label $Y$ of examples within the SMNLI dataset, induced through the described sub-sampling process of the MNLI dataset.

The graphical model described by the DGP for producing the SMNLI dataset is given in Figure 11. Once again, this graphical model can be represented functionally as:

$$S = f_S(U_S); \tag{42}$$
$$C = f_C(S, U_C); \tag{43}$$
$$X = f_X(C, E, U_X); \tag{44}$$
$$Y = f_Y(C, U_Y). \tag{45}$$

More specifically, given the orignal dataset $\mathcal{D}$ that we are sub-sampling from, the functions that we use within the DGP for the SMNLI dataset are given by:

$$S \sim \texttt{Cat}(\mathcal{S}), \tag{46}$$
$$U_C \sim \texttt{Ber}(\rho_{\text{spurious}}); \tag{47}$$
$$C \sim \begin{cases} \hat{C}(S) & \text{if } U_C = 1; \\ R \sim U(\mathcal{C} \setminus \hat{C}(S)) & \text{if } U_C = 0; \end{cases} \text{where } \mathcal{C} = \{0, 1, 2\} \text{ is the entailment label set;} \tag{48}$$
$$X \sim p_{\mathcal{D}}(\cdot|C, S) \tag{49}$$
$$Y = S. \tag{50}$$

Here, $\texttt{Cat}(\mathcal{S})$ is a uniform categorical distribution over spurious feature values (here the underlying genre of the premise-hypothesis pairs). Furthermore, we define $\hat{C}(S)$ to be the entailment label that a particular value of $S$ is spuriously correlated with by design. Moreover, $p_{\mathcal{D}}(\cdot|C, S)$ is the conditional distribution over the dataset examples (premise-hypothesis pairs) that have NLI relationship $C$ and genre $S$.

We restrict the genres within the model to $S \in \{slate, government, fiction, travel\}$, a subset of the genres of the training set. When creating a distribution shifted test set, we restrict the genres to $S \in \{facetoface, nineeleven, verbatim\}$. The specific spurious correlations between genres and labels $Y$ are chosen to be:

1. $\hat{C}(slate) = 0$;

2. $\hat{C}(government) = 2$;

3. $\hat{C}(fiction) = 1$;

4. $\hat{C}(travel) = 0$;

5. $\hat{C}(facetoface) = 2$;

6. $\hat{C}(nineeleven) = 0$;

7. $\hat{C}(verbatim) = 1$.

In this way we generate spurious correlations within the dataset through sub-sampling to induce spurious correlations between $S$ and $Y$. In this case, for a dataset example $X$, the spurious correlation

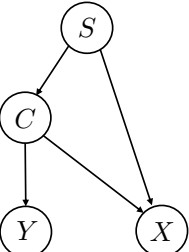

Figure 13: **SHANS DGP**. Graphical model showing the data generating process for modifying examples from the SHANS dataset to introduce a new spurious features $S_{\text{lex.}}$, $S_{\text{sub.}}$ and $S_{\text{const.}}$ which are encoded within the categorical spurious feature $S$ which represents one of these three heuristics.

between $S = s$ and $\hat{C}(s)$ is given as

$$\mathbb{P}(Y = \hat{C}(s)|X, S = s, s \in X) = \rho_{\text{spurious}}. \tag{51}$$

This follows immediately form the sub-sampling process described above. This then once again aligns with the definition of $\rho_{\text{spurious}}$ in Equation (4) and justifies the choice of notion in the above generative process.

**SCM for SMNLI.** The DGP again induces a SCM that induces spurious correlations between spurious features $S$ and the label $Y$. The SCM has the same structure as in the SS dataset, and is given in Figure 14 where once again, $U$ again is some hidden confounding variable.

A.9    SPURIOUS HANS DATASET (SHANS)

We generate a binary NLI dataset, SHANS, with a known spurious feature. We do this considering the HANS dataset McCoy (2019). This is an NLI data set with two labels: entailment (0) and contradiction (1). This is an adversarial dataset designed to assess different NLI models' reliance on spurious heuristics rather than on the underlying relationship between the premise and the hypothesis when making predictions. Specifically, the author's consider three major categories of heuristics: lexical overlap heuristic (assuming that a premise entails from words within the hypothesis) , sub-sequence heuristic( assuming that the premise entails all any of its contiguous sub-sequences of words) and constituent heuristic (assuming that a premise entails a hypothesis that is any constituent within it's syntactic parse tree).

**Data-generating process (DGP).** We consider a graphical model to describe the DGP of examples within the SHANS dataset. We use the following variables within our DGP:

- $C$ - NLI relationship between a premise and hypothesis pair, the causal feature within this task, sampled from the original dataset.
- $S_{\text{lex.}}$ - spurious feature, here the presence of a hypothesis entirely made from words from the premise. This is a binary categorical variable (present/ not present).
- $S_{\text{sub.}}$ - spurious feature, here the presence of a hypothesis that is a contiguous subsequence of the premise. This is a binary category feature (present/ not present).
- $S_{\text{const.}}$ - spurious feature, here the presence of hypothesis that is a constituent/subtree of the premise. Here we have a binary variable (present/ not present).
- $X$ - example from the HANS dataset.
- $Y$ - final label for element $X$.

The graphical model described by the DGP for producing the S-HANS dataset is given in Figure 13. Once again, this graphical model can be represented functionally as

$$S = f_S(U_S); \tag{52}$$
$$C = f_C(S, U_C); \tag{53}$$
$$X = f_X(C, E, U_X); \tag{54}$$
$$Y = f_Y(C, U_Y), \tag{55}$$

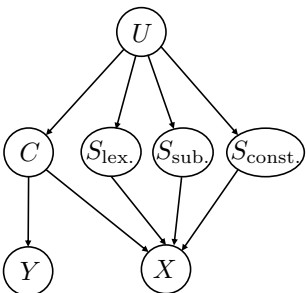

Figure 14: **SHANS SCM.** SCM showing the spurious correlations present between the binary pre-cence of heurstics features $S_{\text{lex.}}$, $S_{\text{sub.}}$ and $S_{\text{const.}}$ and the label $Y$ of examples within the S-HANS dataset, induced through the described sub-sampling process of the S-HANS dataset.

where here we define $S$ to be a categorical feature over the set of the presence of each of the three heuristics introduced above which we denote, through overloaded notation, by $\mathcal{S} = \{S_{\text{lex.}}, S_{\text{sub.}}, S_{\text{const.}}\}$. More specifically, given the original dataset $\mathcal{D}$ that we are sub-sampling from, the functions that we use within the DGP for the S-HANS dataset are given by:

$$S \sim \text{Cat}(\mathcal{S}), \tag{56}$$
$$U_C \sim \text{Ber}(\rho_{\text{spurious}}); \tag{57}$$
$$C \sim \begin{cases} \hat{C}(S) & \text{if } U_C = 1; \\ R \sim U(\mathcal{C} \setminus \hat{C}(S)) & \text{if } U_C = 0; \end{cases} \quad \text{where } \mathcal{C} = \{0, 1\} \text{ is the entailment label set;} \tag{58}$$
$$X \sim p_{\mathcal{D}}(\cdot | C, S) \tag{59}$$
$$Y = S. \tag{60}$$

Here, $\text{cat}(\mathcal{S})$ is a uniform categorical distribution over $\mathcal{S}$ which effectively selects the precence of exactly one of the three spurious feature heuristics. We define $\hat{C}(S)$ to be the entailment label that a particular value of $S$ is spuriously correlated with by design. Moreover, $p_{\mathcal{D}}(\cdot | C, S)$ is the conditional distribution over the dataset examples (premise-hypothesis pairs) that have NLI relationship $C$ and the presence of spurious heuristics $S$.

We consider the presence of each feature to be separate binary spurious features. The specific spurious correlations between heuristics and labels $Y$ are chosen to be:

1. $\hat{C}(S_{\text{lex.}}) = 0$;

2. $\hat{C}(S_{\text{sub.}}) = 0$;

3. $\hat{C}(S_{\text{const.}}) = 1$;

In this way we generate spurious correlations within the dataset through sub-sampling to induce spurious correlations between the heuristics and $Y$. In this case, for a dataset example $X$, the spurious correlation between the presence of a particular heuristic $S = s$ and $\hat{C}(s)$ is given as

$$\mathbb{P}(Y = \hat{C}(s) | X, S = s, s \in X) = \rho_{\text{spurious}}. \tag{61}$$

This follows immediately form the sub-sampling process described above. Once again, this aligns with the definition in Equation (4) which again justifies the introduced notation.

**SCM for SHANS.** The DGP again induces a SCM. In particular, considering $S$ as consiting of three binary spurious features $S_{\text{lex.}}$, $S_{\text{sub.}}$ and $S_{\text{const.}}$. The SCM has a similar structure to as in the SS and S-MNLI datasets, and is given in Figure 14 where once again, $U$ again is some hidden confounding variable.

A.10    FIT ON SHANS

Here we give the results of performing SFT and FIT on the SHANS datasets.

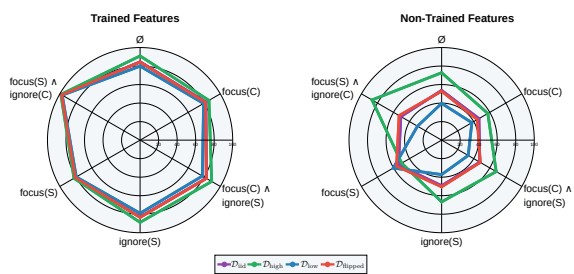

Figure 15: **SHANS focus accuracies** (↑). Focus accuracy ($\mathcal{A}_{\text{focus}}$) of `Llama-3.1-8B-Instruct` after FIT on the SHANS dataset. Here, $C$ refers to the causal feature (logical relationship between premise and hypothesis) and $S$ the spurious feature (heuristic used)

**Spurious HANS (SHANS) dataset.** We generate binary NLI dataset sub-sampled from HANS (Mc-Coy, 2019), a dataset designed to challenge NLI models by exposing common heuristics they rely on, such as lexical overlap (whether the hypothesis shares many words with the premise), sub-sequence (whether the hypothesis is a contiguous sub-sequence of the premise), and constituent (whether the hypothesis is a grammatical sub-structure of the premise). The presence of these heuristics are spuriously correlated with labels through sub-sampling the presence of each of the heuristics from the original dataset. The degree of co-occurrence is governed by $\rho_{\text{spurious}}$, which varies according to the test sets described in Section 3. We ensure that $\rho_{\text{spurious}}$ is the same for all feature values within each dataset split. In particular, we set $\rho_{\text{spurious}}$ to be $0.5, 0.5, 0.9, 0.25$ and $0.9$ on $\mathcal{D}_{\text{train}}, \mathcal{D}_{\text{iid}}, \mathcal{D}_{\text{high}}, \mathcal{D}_{\text{low}}$ and $\mathcal{D}_{\text{flipped}}$ respectively.

**Results.** Figure 15 shows the focus accuracy results of performing SFT and FIT on the SHANS dataset for the `Llama-3.1-8B-Instruct` model. As expected, the trained features show high focus accuracy. However, for non-trained features, we observe lower focus accuracy. This could be attributed to the overlapping nature of the heuristics in SHANS, which are often graded versions of each other with different levels of specificity. For instance, the sub-sequence heuristic can overlap with both lexical overlap and constituent heuristics (e.g., the example with Premise:"Before the actor slept, the senator ran" and Hypothesis: "The actor slept." satisfies all three heuristics). This overlap likely confuses the model during generalisation, as it struggles to distinguish between heuristics not seen during training and those that are similar. These results suggest a potential limitation of FIT when dealing with features that are not sufficiently distinct or have significant overlap.

