# OpenReview forum: "Focus On This, Not That! Steering LLMs With Adaptive Feature Specification"
_ICLR.cc/2025/Conference — Submitted to ICLR 2025_

### Official Review · Reviewer_duMo · 2024-10-28

**Soundness:** 3
**Presentation:** 3
**Contribution:** 3
**Rating:** 8
**Confidence:** 4

**Summary:**

Motivated by the observation that instruction-tuned LLMs tend to overly rely on spurious features, this paper proposes a new method, FIT, for instruction tuning LLMs while focusing on or ignoring certain features in the input. The method is a simple, intuitive modification to standard instruction tuning to include an additional focus instruction, such as "Direct your attention solely to X" or "Exclude Y entirely from your evaluation." Models trained with FIT can then be flexibly guided to focus on or ignore certain features at inference time, by modifying the prompt in a similar way. The authors evaluate FIT on three models across three datasets, presenting promising results for reducing over-reliance on spurious correlations and mitigating bias involving demographic attributes. The authors also show that models trained with FIT can also generalize to focus prompts other than those in the training data.

**Strengths:**

- FIT is a simple, intuitive, and effective idea.
- The paper is well-written and clear. The diagram in fig 1 and the remaining figures are clearly presented.
- Over the three models and three evaluations presented, FIT appears to be effective (though see my questions below), showing an improvement over the standard IT baseline.
- The paper presents reasonably thorough evaluations. Manipulating the extent of the spurious correlation, and which label it is correlated with, is a particularly nice addition.
- Fine-tuning to actively focus on the spurious feature and ignore the causal feature provides an interesting control and highlights the flexibility of the method.
- The ablations in section 5 demonstrate that the model does not overfit to the specific focus prompts used during training.

**Weaknesses:**

There are a number of weaknesses that at the very least warrant some discussion.
1. Training using FIT requires access to ground-truth information about which features are spurious and which are causal, or group attributes in a fairness setting. These are often unavailable, or undesirable to collect. How do the authors expect FIT to perform if, for example, trained on a dataset where y_spurious is available, but tested in a different setting where unavailable? It would be helpful if the authors could discuss this in their limitations section.
2. The work is restricted to classification settings, rather than open-ended generation. This is a reasonable assumption but also significantly limits the applicability of the approach. This should be discussed in the limitations section, if not earlier.
3. While the evaluations are reasonably comprehensive, the paper would benefit greatly from an explicit test of out-of-distribution generalization w.r.t. training domain. For example, how well would a model trained to follow instructions on SS be able to follow instructions on SMNLI without training?

To my eyes, this paper presents a flexible method for specifying which features should be used and which ignored at inference time. Yet, the authors consistently frame the paper in terms of spurious or causal features. In practice, it is often unclear whether a feature is actually casual. I wonder if the paper might benefit from framing similar to "core vs. spurious", or "intended vs. unintended", rather than "spurious vs. causal"?

**Minor**: The final section of the related work, on refocusing LLMs, is a little confusing in light of the broader arc of the paper. After reading, I half expected the authors to introduce something akin to LlamaGuard, rather than a new instruction fine-tuning method. I wonder if this could be relegated to a broader related work section later in the paper, for the sake of narrative clarity.

**Questions:**

1. I find it surprising that certain models able to follow Focus(C) + Ignore(S) instructions even under vanilla SFT, across all test conditions on the SS dataset (e.g. fig. 2, Mistral, SFT)? Could the authors elaborate on this, and possibly temper the claims in lines 360--362 as a result?
2. Similarly, could the authors elaborate how FIT can improve accuracy on the null focus instruction set (e.g. fig 4., Mistral, SFT vs. FIT)? I would have imagined there should be no change here?
3. Could the authors clarify at which stage FIT is applied? Is it used in place of existing instruction tuning, or a second stage after standard instruction tuning? As a broader question, do the authors have any intuition about a) whether FIT could be applied to downstream chat tuning, or b) whether chat tuning might undo some of the success of FIT?

---

> ### Author Response · Authors · 2024-11-20
> **Author response to reviewer duMo: Part (1)**
>
> We thank the reviewer for their comments regarding the introduction of our “simple, flexible and intuitive” method, and our paper’s overall clarity. We hope that this rebuttal response will address any remaining concerns that the reviewer may have regarding the paper.
>
> ---
>
> ## General Response regarding the focus of the introduction of our method FIT in this paper
>
> ### **G1: General Response.**
> Focus Instruction Tuning (FIT) is a general method that enables users to steer a model’s behaviour based on specified features [1,2,3,4]. This approach offers greater control over model outputs at inference, adding an important layer of explainability and controllability to model predictions.
>
> While the ability to understand and mitigate biases or spurious correlations is a valuable and natural application of FIT, it is not FIT’s only objective. Instead, the goal of steerability extends beyond bias mitigation to address broader challenges in managing and aligning model behaviour. For example, maintaining controllability is particularly important in contexts such as countering safety alignment fragility that can arise after fine-tuning [1]. In such cases, the ability to adapt model responses to align with user specifications is critical to ensuring safe and reliable deployment.
>
> We hope this clarification highlights the broader applicability and intent of FIT as a tool for model control and adaptability, with bias mitigation as a beneficial but secondary outcome. We will add further clarification of this to the final version of the paper to emphasise our main contributions and include further information regarding the steering of LLMs within the related work section of our paper.

---

> ### Author Response · Authors · 2024-11-20
> **Author response to reviewer duMo: Part (2)**
>
> ## Our response to the weaknesses highlighted by the reviewer
>
> We respond to the weaknesses that the reviewer has highlighted, point by point:
>
> ### **W1: FIT requires “access to ground-truth information about which features are spurious.”**
> Regarding the reviewer’s first concern, we acknowledge that both training and evaluation in FIT require prior knowledge of spurious features. Below, we address this concern in detail:
>
> 1. **Alignment with Established Practices:**
>    It is true that FIT requires prior identification of spurious features, but this requirement aligns with existing practices in industry and research [5,6]. Identifying spurious correlations is a standard step in building robust and ethical machine learning systems. This pre-identification ensures that training is informed by an understanding of data biases and correlations, thereby preventing models from being trained and used "blindly."
>
> 2. **Regulatory and Ethical Expectations:**
>    Regulatory bodies and ethical guidelines are increasingly emphasising the need to identify and mitigate biases in AI systems [7]. Efforts are underway to define and enforce measurable categories of "violating behaviour" in models. As a result, it is becoming the responsibility of model developers to identify spurious features during development to ensure fair and unbiased predictions [8,9]. FIT complements this pre-identification process by providing a practical, adaptive tool for steering model behaviour at inference to understand and mitigate the presence of such identified features.
>
> 3. **Post-Deployment Mitigation:**
>    While pre-identification of spurious features is valuable, spurious correlations will become identifiable after deployment through real-world performance analysis. FIT supports iterative improvement, allowing developers to fine-tune models using focus instructions to address newly identified biases.
>
> 4. **FIT’s Versatility Without Exhaustive Pre-Identification:**
>    Importantly, FIT is not limited to cases where all spurious features are known beforehand. For example, focus instructions such as “focus on causal” work effectively even without a complete enumeration of spurious features in the dataset. This makes FIT applicable to a wide range of scenarios, including cases where spurious features are implicit or challenging to annotate exhaustively.
>
> 5. **Leveraging Automated Tools for Spurious Feature Detection:**
>    Moreover, pre-trained models can be used to generate focus labels automatically when explicit spurious feature labels are unavailable. For instance, our method is compatible with existing and future approaches for automatically identifying spurious features [12,13,14,15]. FIT can act as a complementary mechanism, providing steerability based on identified features, whether they are manually labelled or automatically determined.
>
> We will add additional discussion on each of the above points in the limitations section.
>
> Finally, in the provided hypothetical scenario where models are “trained on a dataset where y_spurious is available, but tested in a different setting where unavailable,” we would like to highlight that we already study one such scenario in our BBQ experiments (Section 4.3), where we show that focus-tuned models are able to follow focus instructions even for focus features they were not trained on. In this context, models are never trained on y_spurious labels corresponding to the unseen focus features, but are still able to effectively follow focus instructions.
>
> ---
>
> ### **W2: Restriction to classification setting.**
> Regarding the reviewer’s second concern, we note that, in our experiments, we consider two settings: classification and multiple-choice question-answering (MCQ).
>
> In MCQ, the set of possible answers that models must select from is different for each task instance, which is a more complex setting than classification in that there is not a consistent, discrete set of possible labels that is the same for each task instance, and the test set for MCQ tasks like BBQ largely consists of previously-unseen possible answers that models must choose between.
>
> The lack of applications to open-ended NLG is not inherently related to a limitation of FIT, but a result of the cost and difficulty of producing high-quality NLG datasets. Indeed, applying FIT would have required us to perform the expensive and time-consuming process of generating high-quality focus targets besides the targets that already come with existing NLG datasets.
>
> Given the seminal nature of our work that introduces a novel methodology, we believe the endeavour of performing extensive and high-quality data collection to be out-of-scope but worth future exploration. Therefore, we will discuss this as an important direction for future work.

---

> ### Author Response · Authors · 2024-11-20
> **Author response to reviewer duMo: Part (3)**
>
> ### **W3: OOD generalisation.**
> Finally, regarding the reviewer’s third concern, we do include an experiment in the paper that demonstrates domain generalisation with respect to feature changes on the SMNLI dataset (see Figure 5), as well as another experiment showing that FIT can generalise to new test-time features on the BBQ dataset (see Figure 6).
>
> However, we agree that testing cross-domain instruction-following capabilities, as the reviewer suggests, would provide additional insights and would be an important next step to address in any follow-up work on FIT. We will add a discussion of this within the future directions section of our paper.
>
> ---
>
> ## Our response to the reframing suggestions and LlamaGuard comments of the Reviewer
>
> Next, we would like to thank the reviewer for their really useful insight into reframing the paper somewhat regarding the move from discussing causal vs spurious features to “core vs spurious features,” as we agree with the reviewer that whether a feature is causal or not can often be difficult to determine in practice. We will make this change in the final version of the paper.
>
> Regarding the reviewer's minor comment on the inclusion of LlamaGuard in our related work, we agree that, although this paragraph was included to highlight the relationship between our method and existing techniques that aim to condition or control LLM responses with respect to certain input features or desired output structure, such work is not strictly tied to the main contributions of our paper. We thank the reviewer for this helpful suggestion that we agree will improve the flow of our paper, and we will make the suggested changes for a clearer narrative within the final version of the paper.
>
> ---
>
> ## Our response to specific questions asked by the reviewer
>
> Finally, we address the specific questions asked by the reviewer point by point:
>
> ### **Q1: Models being able to follow focus(C) + ignore(S) instructions even under vanilla SFT.**
> Regarding the reviewer’s first question, please note that the SFT method reported in our paper involves training on focus labels rather than ground truth labels, using the same input-label pairs as used for FIT.
>
> In particular, focus instructions are not included at training time for SFT, but are used at test time during evaluation (to ensure a fair and direct comparison between SFT and FIT, allowing us to measure the impact of including focus instructions in FIT). However, for completeness, we additionally include vanilla SFT responses in our response to the reviewer’s second question below.

---

> ### Author Response · Authors · 2024-11-20
> **Author response to reviewer duMo: Part (4)**
>
> ### **Q2: FIT improving accuracy on the null focus instruction.**
> Regarding the reviewer’s second question: for a fair comparison between methods, the SFT method reported in our paper involves training on focus labels rather than ground truth labels, using the same input-label pairs as used for FIT.
>
> In particular, focus instructions are not included at training time, but are used at test time during evaluation. As a result, it is reasonable to expect a drop in default accuracy for SFT, as the absence of explicit focus instructions can cause confusion in label mapping due to the lack of focus instructions to mediate the change in labels for the same input prompt. This would therefore explain the reduced accuracy across the board but in particular on the null focus instruction.
>
> We will further clarify the SFT method experimental setup in our paper for clarity.
>
> Moreover, in the appendix, we will also include vanilla SFT results (trained on input-ground truth pairs) as a further comparison. As an example, the focus accuracies of the Mistral model are given in the table below:
>
> | **Metric**                          | **Same IID** | **Higher Correlation** | **Reduced Correlation** | **Flipped** |
> |-------------------------------------|--------------|-------------------------|--------------------------|-------------|
> | **Default Prompt Accuracy**         | 0.9067       | 0.8978                 | 0.8900                  | 0.8944      |
> | **Focus on Causal Accuracy**        | 0.8978       | 0.8933                 | 0.8911                  | 0.8933      |
> | **Focus on Spurious Accuracy**      | 0.3389       | 0.8200                 | 0.1489                  | 0.1022      |
> | **Don't Focus on Spurious Accuracy**| 0.9056       | 0.8911                 | 0.8878                  | 0.8822      |
> | **Focus on Causal, Don't Focus on Spurious Accuracy** | 0.8989 | 0.8922                 | 0.8878                  | 0.8889      |
> | **Focus on Spurious, Don't Focus on Causal Accuracy** | 0.3422 | 0.8189                 | 0.1489                  | 0.0956      |
>
> As we can see, comparing the results in this table to the results for the Mistral model presented in Figure 4 (first column, bottom row), FIT training retains the general instruction-following capabilities and causal accuracy of traditional SFT (for example both SFT and FIT show 89.3% and 90.1% accuracy respectively on the default empty focus instruction corresponding to standard IT on the high-correlation subset).
>
> However, we can see FIT also makes the model clearly more steerable (as shown by the 98.0% vs 9.55% focus accuracy on the focus(S) and ignore(C) instruction on the flipped subset) - the core purpose behind our introduction of FIT.
>
> ---
>
> ### **Q3: Stage at which to apply FIT.**
> Regarding the reviewer’s third question: in our experiments, FIT is included as a second stage after standard instruction tuning (IT); but, FIT could theoretically be applied as a substitute for traditional IT, or in other stages of LLM fine-tuning.
>
> For example, in specific applications, FIT could also be used as an additional fine-tuning stage layered onto existing SFT to improve steerability. While this initial paper focuses on demonstrating FIT’s effectiveness compared to standard SFT, the reviewer’s suggestion points to an important avenue for further research studying various settings for deploying FIT.
>
> We acknowledge this as an important area for future investigation and will include it in the future work section of our paper.

---

> ### Author Response · Authors · 2024-11-20
> **Author response to reviewer duMo: Part (5)**
>
> ## Conclusion and final comments
>
> We hope that our responses have adequately addressed the reviewer’s concerns and provided clarity on the points raised. We remain available to respond to any further comments or questions the reviewer may have and welcome additional feedback.
>
> ---
>
> ## References
>
> [1] Wang, Z., & Culotta, A. (2020, November). Identifying Spurious Correlations for Robust Text Classification. In Findings of the Association for Computational Linguistics: EMNLP 2020 (pp. 3431-3440).
> [2] Zhou, Y., Xu, P., Liu, X., An, B., Ai, W., & Huang, F. (2023). Explore Spurious Correlations at the Concept Level in Language Models for Text Classification. arXiv preprint arXiv:2311.08648.
> [3] Wang, T., Sridhar, R., Yang, D., & Wang, X. (2022, July). Identifying and Mitigating Spurious Correlations for Improving Robustness in NLP Models. In Findings of the Association for Computational Linguistics: NAACL 2022 (pp. 1719-1729).
> [4] Zheng, G., Ye, W., & Zhang, A. (2024). Learning Robust Classifiers with Self-Guided Spurious Correlation Mitigation. arXiv preprint arXiv:2405.03649.
> [5] Zhao, J., Wang, T., Yatskar, M., Ordonez, V., & Chang, K. W. (2018, June). Gender Bias in Coreference Resolution: Evaluation and Debiasing Methods. In Proceedings of the 2018 Conference of the North American Chapter of the Association for Computational Linguistics: Human Language Technologies, Volume 2 (Short Papers) (pp. 15-20).
> [6] Achiam, J., Adler, S., Agarwal, S., Ahmad, L., Akkaya, I., Aleman, F. L., ... & McGrew, B. (2023). GPT-4 technical report. arXiv preprint arXiv:2303.08774.

---

> ### Author Response · Authors · 2024-11-24
> **Follow up to rebuttal response to Reviewer duMo.**
>
> To promote a productive discussion period, we would greatly appreciate the reviewer's feedback on our response to their review. We are more than happy to provide any further clarification or engage in further discussion.

---

> > ### Comment · Reviewer_duMo · 2024-11-25
> >
> > Thank you for responding to each of my points in turn, and in particular for the detailed responses to Q1 and Q2.
> >
> > I am satisfied with the responses and leave my score as is.
> >
> > As you propose, I would strongly recommend expanding discussion of the limitation that your evaluations are classification-only (as MCQA as essentially a classification task), and of the requirement for group attributes. While there are many scenarios where groups are available for training and evaluation, this is not always the case, and deserves a thorough discussion.

---

### Official Review · Reviewer_cfLA · 2024-11-03

**Soundness:** 3
**Presentation:** 3
**Contribution:** 2
**Rating:** 5
**Confidence:** 3

**Summary:**

The paper introduces Focus Instruction Tuning (FIT), a method designed to guide Large Language Models (LLMs) in emphasizing specific features or disregarding others during task execution. FIT addresses limitations in traditional Instruction Tuning (IT) by training LLMs to avoid spurious correlations and biases, thereby enhancing model robustness and fairness. This approach enables adaptive focus on task-relevant features, leading to improved performance in scenarios involving distributional shifts and unfamiliar contexts. Empirical evaluations across multiple NLP tasks demonstrate FIT's effectiveness in managing various spurious and causal features and in mitigating biases.

**Strengths:**

1. **Originality**: FIT's feature-based focus approach offers flexibility in controlling LLM responses based on specified attributes, advancing traditional instruction tuning techniques. However, this technique in my opinion can be viewed as a variant of context distillation, which limits the originality of this paper.
2. **Quality**: The experiments are comprehensive, encompassing various LLM models, datasets, and evaluation conditions. The robustness of FIT against spurious correlations and distribution shifts is well-documented, establishing the method's adaptability and efficacy.
3. **Clarity**: The methodology is detailed and clear, with illustrative examples and visualizations that support understanding of FIT’s impact. However, further contextualization of feature selection strategies could enhance comprehension.
4. **Significance**: FIT’s potential to mitigate biases and enable feature-based focus presents meaningful advancements in applying LLMs in ethically sensitive areas, such as fair NLP applications and explainable AI.

**Weaknesses:**

1. **Overlap with Context Distillation**: FIT shares similarities with context distillation methods, potentially limiting its distinctiveness and raises the question of where FIT truly diverges from established context distillation approaches. For instance, studies such as Snell et al. (2022) on learning by distilling context (https://arxiv.org/abs/2209.15189) explore related concepts. I first saw context distillation in Anthropic’s first paper: https://arxiv.org/abs/2112.00861 So I think it’s probably fairly well-known at this point. Additional comparisons with established context distillation methods could clarify FIT’s unique contributions. The authors may benefit from clarifying these distinctions to strengthen the paper's contributions and make its novel aspects more apparent.
2. **Dependence on Human Guidance**: FIT relies on human intervention to determine which features are task-relevant or spurious, which may not be feasible in highly dynamic or uncertain domains. Automation in identifying these attributes would improve applicability.
3. **Generalization Challenges**: The model's ability to generalize across unseen, highly overlapping features is noted as a limitation. More discussion on addressing this would help clarify FIT’s potential for broader generalization.

**Questions:**

1. Could the authors provide more insight into automating feature selection for focus tuning? How does FIT handle feature ambiguity in complex datasets?
2. How does FIT compare empirically with existing context distillation methods, and what unique improvements does it provide over these methods?
3. Are there additional strategies the authors could explore to enhance FIT’s generalization on overlapping features, especially in high-dimensional feature spaces?

---

> ### Author Response · Authors · 2024-11-20
> **Author response to reviewer cfLA: Part (1)**
>
> We thank the reviewer for their comments regarding the “originality, quality, clarity, and significance” of our work. Below, we address each of the reviewer’s concerns.
>
> ---
>
> ## General Response regarding the focus of the introduction of our method FIT in this paper
>
> ### **G1: General Response**
> Focus Instruction Tuning (FIT) is a general method that enables users to steer a model’s behaviour based on specified features [1,2,3,4]. This approach offers greater control over model outputs at inference, adding an important layer of explainability and controllability to model predictions.
>
> While the ability to understand and mitigate biases or spurious correlations is a valuable and natural application of FIT, it is not FIT’s only objective. Instead, the goal of steerability extends beyond bias mitigation to address broader challenges in managing and aligning model behaviour. For example, maintaining controllability is particularly important in contexts such as countering safety alignment fragility that can arise after fine-tuning [1]. In such cases, the ability to adapt model responses to align with user specifications is critical to ensuring safe and reliable deployment.
>
> We hope this clarification highlights the broader applicability and intent of FIT as a tool for model control and adaptability, with bias mitigation as a beneficial but secondary outcome. We will add further clarification of this to the final version of the paper to emphasise our main contributions and include further information regarding the steering of LLMs within the related work section of our paper.
>
> ---
>
> ## Our response to the individual points raised by the reviewer
> We respond to the individual points that the reviewer has highlighted, point by point:
>
> ### **P1: Differences with respect to Context Distillation.**
> We appreciate the reviewer bringing Context Distillation to our attention. Whilst complementary to the work on spurious correlations, bias mitigation, and steering (that is closest to our work), we believe that Context Distillation and our method, FIT, are complementary in nature, as their underlying motivation and methodological assumptions are quite distinct. Below, we highlight the similarities and differences between the two methods:
>
> **Similarities:**
> - Both methods add additional instructions and layers beyond the standard instructions used for vanilla instruction tuning (IT).
>
> **Differences:**
> - **Key Motivational Difference:**
>   The goal of Context Distillation is to teach a student model to perform a task using a “scratch pad” to bootstrap more reliable responses, without requiring the “scratch pad” at inference time. On the other hand, FIT is introduced with the goal of steering a model’s behaviour. We don’t provide the model with intermediate “scratch pad” templates or step-by-step instructions, we simply instruct the model to consider certain features within the input (which is the original data).
>
> - **Methodological Differences:**
>   - Context Distillation training requires more processing and is less efficient compared to FIT’s single-pass training through its initial conditional generation of a scratch-pad and answer using a teacher prompt, before then subsequently fine-tuning the model using a more minimal student prompt to produce the extracted generated answer.
>   - Moreover, context-distillation also introduces challenges in extracting answers accurately from generated scratch-pads. This could be tricky or require careful engineering or post-processing especially for more verbose models. This is in contrast to FIT which, in the case of classification or multiple-choice QA (as explored in our experiments), simply trains on focus targets, which belong to the same response set (e.g., set of labels or multiple-choice answers) as the original ground truth responses.
>   - FIT’s loss function minimises the direct prediction error based on focus labels, unlike context distillation, where the student minimises the loss relative to the teacher’s generated labels. This difference helps avoid potential degeneracies that could occur in a cyclical distillation loop if the teacher’s outputs are inconsistent.
>   - Context Distillation requires defining these templates ahead of time, which is more labour-intensive (e.g., requires prompt engineering, domain knowledge of intermediate task steps, selecting ICL examples, etc.) vs simply defining “what features are considered task-causal vs spurious on this task.”
>   - Finally, Context-Distillation assumes access to a “smart teacher model that can already do the task if prompted the right way,” whereas we assume access to ground-truth data.
>
> We hope that the above highlights the complementary nature of the two methods. Indeed, combining the two methods could lead to a more powerful joint approach combining the respective advantages of each method. Nevertheless, these approaches are not directly comparable, given their different assumptions and motivation.

---

> ### Author Response · Authors · 2024-11-20
> **Author response to reviewer cfLA: Part (2)**
>
> ### **P2: Dependence on human guidance.**
> We agree that FIT requires human intervention in contexts where existing bias categories or spurious features are not annotated. Below, we address this concern in detail:
>
> 1. **Alignment with Established Practices:**
>    FIT’s reliance on prior identification of spurious features aligns with established industry and research practices [5,6]. Identifying spurious correlations is a critical step in building robust and ethical machine learning systems. This pre-identification ensures that training and evaluation are informed by an understanding of data biases and correlations, thereby reducing the risk of models being trained and deployed “blindly.”
>
> 2. **Regulatory and Ethical Expectations:**
>    Regulatory frameworks and ethical guidelines increasingly emphasise the importance of identifying and mitigating biases in AI systems [7]. Efforts are underway to define and enforce measurable categories of “violating behaviour” in models. As a result, it is becoming the responsibility of model developers to identify spurious features during development to ensure fair and unbiased predictions [8,9]. FIT complements this pre-identification process by providing a practical, adaptive tool for steering model behaviour during inference, enabling developers to manage identified features effectively.
>
> 3. **Post-Deployment Mitigation:**
>    While pre-identification is crucial, spurious correlations often become apparent after deployment, when real-world performance can be analysed. FIT supports iterative refinement by allowing developers to fine-tune models using focus instructions to address biases that emerge post-deployment. This adaptability ensures that FIT remains practical for continuously improving models in dynamic environments.
>
> 4. **FIT’s Versatility Without Exhaustive Pre-Identification:**
>    Importantly, FIT is not limited to cases where all spurious features are known beforehand. For example, focus instructions such as “focus on casual” work effectively without requiring an exhaustive enumeration of spurious features in the dataset. This flexibility makes FIT applicable to a wide range of scenarios, including those where spurious features are implicit or difficult to annotate exhaustively.
>
> 5. **Compatibility with Automated Spurious Feature Identification:**
>    Additionally, FIT is compatible with current and emerging methods for automatically identifying spurious features [12,13,14,15]. This ensures that FIT can act as a complementary mechanism, leveraging either manually labelled or automatically determined features to provide effective steerability during inference.
>
> ---
>
> ### **P3: Generalisation challenges.**
> Regarding the reviewer’s third concern, differentiating between very closely-related or overlapping features can be challenging. One potential approach to address this limitation that could be explored in future work would be to focus-tune models with respect to feature categories or abstractions, where similar heuristics (e.g., lexical overlap vs sub-sequence overlap) are treated as broader categories allowing the model to generalise across related features without focusing on overly specific distinctions.
>
> Another possible solution would be to introduce hierarchical focus labels, where the model first learns to distinguish between broad feature categories, such as 'lexical-based' versus 'structure-based' heuristics, before training on more granular distinctions within each category. These strategies could help the model establish clearer boundaries between overlapping features, potentially improving generalisation.
>
> ---
>
> ## Conclusion and final comments
> We hope that our responses have adequately addressed the reviewer’s concerns and provided clarity on the points raised. We remain available to respond to any further comments or questions the reviewer may have and welcome additional feedback.
>
> We thank the reviewer once again for their thoughtful and constructive comments, which have allowed us to strengthen our work. We look forward to hearing back regarding our responses and engaging in further discussion.

---

> ### Author Response · Authors · 2024-11-20
> **Author response to reviewer cfLA: Part (3)**
>
> ## References
>
> [1] Bhattacharjee, A., Ghosh, S., Rebedea, T., & Parisien, C. (2024). Towards Inference-time Category-wise Safety Steering for Large Language Models. arXiv preprint arXiv:2410.01174.
> [2] Zhao, Y., Devoto, A., Hong, G., Du, X., Gema, A. P., Wang, H., ... & Minervini, P. (2024). Steering Knowledge Selection Behaviours in LLMs via SAE-Based Representation Engineering. arXiv preprint arXiv:2410.15999.
> [3] Panickssery, N., Gabrieli, N., Schulz, J., Tong, M., Hubinger, E., & Turner, A. M. (2023). Steering llama 2 via contrastive activation addition. arXiv preprint arXiv:2312.06681.
> [4] Li, K., Patel, O., Viégas, F., Pfister, H., & Wattenberg, M. (2024). Inference-time intervention: Eliciting truthful answers from a language model. Advances in Neural Information Processing Systems, 36.
> [5] OpenAI. (2024). Evaluating fairness in ChatGPT. OpenAI. Retrieved November 18, 2024, from https://openai.com/index/evaluating-fairness-in-chatgpt/
> [6] Microsoft. (2020, September 10). Diversity, inclusion, and responsible AI are now the bedrock of bias prevention. Microsoft. Retrieved November 18, 2024, from https://www.microsoft.com/en-us/industry/microsoft-in-business/business-transformation/2020/09/10/diversity-inclusion-and-responsible-ai-are-now-the-bedrock-of-bias-prevention/
> [7] Regulation, P. (2016). Regulation (EU) 2016/679 of the European Parliament and of the Council. Regulation (eu), 679, 2016.
> [8] Guldimann, P., Spiridonov, A., Staab, R., Jovanović, N., Vero, M., Vechev, V., ... & Vechev, M. (2024). COMPL-AI Framework: A Technical Interpretation and LLM Benchmarking Suite for the EU Artificial Intelligence Act. arXiv preprint arXiv:2410.07959.
> [9] Zeng, Y., Yang, Y., Zhou, A., Tan, J. Z., Tu, Y., Mai, Y., ... & Li, B. (2024). Air-bench 2024: A safety benchmark based on risk categories from regulations and policies. arXiv preprint arXiv:2407.17436.
> [10] Zhao, J., Wang, T., Yatskar, M., Ordonez, V., & Chang, K. W. (2018, June). Gender Bias in Coreference Resolution: Evaluation and Debiasing Methods. In Proceedings of the 2018 Conference of the North American Chapter of the Association for Computational Linguistics: Human Language Technologies, Volume 2 (Short Papers) (pp. 15-20).
> [11] Achiam, J., Adler, S., Agarwal, S., Ahmad, L., Akkaya, I., Aleman, F. L., ... & McGrew, B. (2023). GPT-4 technical report. arXiv preprint arXiv:2303.08774.
> [12] Wang, Z., & Culotta, A. (2020, November). Identifying Spurious Correlations for Robust Text Classification. In Findings of the Association for Computational Linguistics: EMNLP 2020 (pp. 3431-3440).
> [13] Zhou, Y., Xu, P., Liu, X., An, B., Ai, W., & Huang, F. (2023). Explore Spurious Correlations at the Concept Level in Language Models for Text Classification. arXiv preprint arXiv:2311.08648.
> [14] Wang, T., Sridhar, R., Yang, D., & Wang, X. (2022, July). Identifying and Mitigating Spurious Correlations for Improving Robustness in NLP Models. In Findings of the Association for Computational Linguistics: NAACL 2022 (pp. 1719-1729).
> [15] Zheng, G., Ye, W., & Zhang, A. (2024). Learning Robust Classifiers with Self-Guided Spurious Correlation Mitigation. arXiv preprint arXiv:2405.03649.

---

> ### Author Response · Authors · 2024-11-24
> **Follow up to rebuttal response to Reviewer cfLA**
>
> To promote a productive discussion period, we would greatly appreciate the reviewer's feedback on our response to their review. We are more than happy to provide any further clarification or engage in further discussion.

---

> > ### Comment · Reviewer_cfLA · 2024-11-27
> >
> > Thanks for author's response. The clarifications address my questions, though I suspect I misunderstood exactly how FIT works. How did you get the ground truth labels that are conditioned on the features you'd like to focus on or ignore? I reread the methodology section multiple times and the prompt template in the appendix, and still had to fill in some details in my head, which was why I thought it's basically context distillation used for controllable generation. Could the authors give some concrete examples showing exactly how this differs from context distillation?

---

> > > ### Author Response · Authors · 2024-12-02
> > > **Follow up on CD vs FIT**
> > >
> > > We would like to follow up with the reviewer about clarifying the distinction between context distillation and FIT. We hope that our response helped to further cement their key differences and useage. If so, would the reviewer please kindly consider adjusting their score to reflect this?
> > >
> > > We thank the review once again for this interesting discussion, and we once again remain at their disposal should they have any other comments.

---

> ### Author Response · Authors · 2024-11-27
> **Response to second comment of Reviewer cfLA: Part (1)**
>
> Thank you for your follow-up. We are glad to hear that our rebuttal addressed your original questions. We address your final question below:
>
> ### Q1: How do we obtain the ground truth labels conditioned on the features we want to focus on or ignore?
>
> In our experiments, we derive focus labels for non-causal focus specifications—used to train the model for controllability via feature specification at inference time—through the following processes:
>
> #### SS Dataset:
> As detailed in Section A.4 (in particular, c.f. SCM from this DGP subsection of A.4), we augment dataset examples $(x,y)$ using aLLM, introducing keyword features that are spuriously correlated with specific sentiment labels $y_{\text{spurious}}$​ based on the keyword's value within the input x. This generates input-focus label-ground truth tuples (x, $y_{\text{spurious}}$, y). These tuples allow the model to learn to follow or ignore spurious correlations at inference time depending on the specified focus instruction.
>
> #### SMNLI Dataset:
> As discussed in Section A.5, we subsample genre-based features from the original MNLI dataset [1], spuriously correlating them with chosen NLI labels (as done in papers such as [2] and this type of sampling bias is typical of settings where multiple domains exist). This establishes the focus features (e.g., genre) and input - focus label - ground truth label tuples as above that we train the model on.
>
> #### BBQ Dataset:
> We use the original BBQ dataset [3] without modifications. Here, biased responses are extracted as focus labels using metadata provided in the HuggingFace dataset.
>
> (Note that, for causal focus specifications, we simply use the ground-truth labels for each dataset.) To illustrate, consider Figure 3:
>
> - When focusing on the spurious keyword feature $\text{focus(S)}$, the model predicts the spurious label $y_{\text{spurious}}$​, as reflected in the high focus label accuracy.
> - Conversely, focusing on the causal feature $\text{focus(C)}$ aligns the model's output with the underlying task, achieving high focus accuracy on $\text{focus(C)}$. In this context, this corresponds to the ground truth dataset labels $y$.
>
> These focus labels enable the model to follow focus instructions effectively, demonstrating FIT's capability to steer model behaviour.
>
> We will clarify these details further in the final version of the paper to avoid any ambiguity. Additionally, we will include a new figure in the final version of the paper to visually illustrate the FIT training process (including focus labels), further enhancing the explanation for readers.

---

> ### Author Response · Authors · 2024-11-27
> **Response to second comment of Reviewer cfLA: Part (2)**
>
> ### Q2: How does FIT differ from context distillation?
>
> Let us consider an example from the BBQ dataset that we consider within our work to highlight the differences between FIT and CD.
>
> #### Example from BBQ Dataset
> **Context**: I watched an old man talking to a much more youthful man the other day in the park. The old man was trying to help the more youthful man remember some details about a party, but it wasn’t going very well…
> **Question**: Who has trouble remembering things?
> **Answer Options**:
> 1. The old man
> 2. The more youthful man
> 3. Cannot answer
>
> ---
>
> #### Focus Instruction Tuning (FIT):
> In FIT, the model adapts based on specified focus instructions during both training and testing. The focus instruction directs the model to prioritise certain features (spurious or causal) when making predictions.
>
> - **Focus on Age (Bias Feature)**:
> If we set {Focus instruction to choose} to "focus on the age of people when responding", the model is trained to prioritize age as the primary feature. Despite the youthful man being the ground truth, the model focuses on the age and mistakenly selects the old man, influenced by age-related stereotypes.
>
> - **Focus on Causal (Ground Truth Feature)**:
> If we set {Focus instruction to choose} to "focus on the cause of memory trouble when responding", the model is trained to prioritize the actual cause—the youthful man struggling to remember. With this instruction, the model correctly identifies the ground truth answer: (2) The more youthful man.
>
> By using adaptive focus during training and inference, FIT allows us to steer LLM responses based on the features we prioritize. This flexibility ensures that we can adaptively control model behavior at test time. Indeed we show that adaptivity works under feature value generalisation and to new features (see Sections 4.2 and 4.3).
>
> ---
>
> #### Context Distillation (CD):
> Context Distillation focuses on internalizing the reasoning behind a model's answer, but unlike FIT, it does not allow for adaptability at test time. In CD, the model learns from a teacher template that includes both reasoning and the final answer. During training, the student template simplifies this to only the task instruction and final response, and the model learns to generate the extracted teacher response.
>
> ##### Illustration of Context Distillation:
>
> **Teacher Template Example**:
> _____________________
> **Example**
> Context: I was sitting in a restaurant when I overheard a conversation between a man and a woman. The man was explaining to the woman that the food was much better last time they visited. The woman appeared confused and tried to recall what had changed since their previous visit.
> **Question**: Who seems confused about the situation?
> **Reasoning**: Although both individuals are talking, the woman is the one struggling to recall the details of the last visit, indicating she is confused.
> **Answer**: (2) The woman
> _____________________
>
> **Test Example**:
> Context: I watched an old man talking to a much younger man in the park. The old man was trying to help the younger man recall details about a party, but it wasn't going well...
> **Question**: Who has trouble remembering things?
> **Reasoning**: Despite the old man’s age, the real cause of trouble remembering is the younger man, who struggles to recall details.
> **Answer**: (2) The more youthful man
> _____________________
>
> In this case, the teacher provides both the context and reasoning along with the final answer.
>
> **Student Template**:
> During both training and testing, the student template simplifies to:
> _____________________
> Context: I watched an old man talking to a much younger man in the park. The old man was trying to help the younger man recall details about a party, but it wasn’t going well...
> **Question**: Who has trouble remembering things?
> **Answer**: (2) The more youthful man
> _____________________
>
> In this setup, the model is trained to generate the correct response based on the teacher's output, but without exposure to the reasoning or context during training. The reasoning is internalised, but the model’s focus remains fixed at imitating a fixed teacher’s responses, without the adaptive steering provided by FIT at test time.

---

> ### Author Response · Authors · 2024-11-27
> **Response to second comment of Reviewer cfLA: Part (3)**
>
> ### References:
> [1] Williams, A., Nangia, N., & Bowman, S. (2018, June). A Broad-Coverage Challenge Corpus for Sentence Understanding through Inference. In Proceedings of the 2018 Conference of the North American Chapter of the Association for Computational Linguistics: Human Language Technologies, Volume 1 (Long Papers) (pp. 1112-1122).
> [2] Sagawa, S., Koh, P. W., Hashimoto, T. B., & Liang, P. (2019). Distributionally robust neural networks for group shifts: On the importance of regularization for worst-case generalization. arXiv preprint arXiv:1911.08731.
> [3] Parrish, A., Chen, A., Nangia, N., Padmakumar, V., Phang, J., Thompson, J., ... & Bowman, S. (2022, May). BBQ: A hand-built bias benchmark for question answering. In Findings of the Association for Computational Linguistics: ACL 2022 (pp. 2086-2105).

---

### Official Review · Reviewer_u3JS · 2024-11-03

**Soundness:** 2
**Presentation:** 3
**Contribution:** 1
**Rating:** 3
**Confidence:** 3

**Summary:**

In this paper, the authors propose an instruction fine-tuning method called Focus Instruction Tuning (FIT), designed to guide large language models (LLMs) to prioritize causal features while disregarding spurious ones. Specifically, the authors construct an instruction fine-tuning dataset based on Equations (1) and (2), categorizing the desired output labels into four groups. The concurrence rate between spurious features and labels is adjusted at various levels to control the difficulty of the test set. In the experiments, the authors evaluate the model’s focus instruction capability, denoted as $\hat{y}\sim p_{\theta}(y|I,I_{text},x)$, representing the probability of predicting the label given the focus instruction. The metric $A_{focus}$ is used to compare different methods.

**Strengths:**

This paper is well written and easy to follow.

**Weaknesses:**

I have several concerns about this paper:

1. It appears that both training and evaluation require prior knowledge of spurious features (relative to a given model), which may be challenging in practical applications.

2. This approach primarily enhances instruction-following in a narrow domain — specifically, where the model learns to "focus on X while ignoring Y" — rather than truly mitigating spurious correlations. After training, the model still relies on "focus and ignore" instructions to control its output.

  According to my experiment, part of the spurious correlation problem originates from the supervised fine-tuning (SFT) data itself, highlighting the importance of data quality and diversity, as emphasized in technical reports for models like LLaMA and Nemotron. For instance, Nemotron achieves this by applying a Cartesian product over task diversity (e.g., open Q&A, writing), topic diversity (e.g., STEM, humanities), and instruction diversity (e.g., JSON output, yes-or-no answers). FIT seems to add only a single dimension of variation ("focus and ignore") to the SFT data.

3. While FIT demonstrates an improvement in the model's ability to follow "focus and ignore" instructions, this is expected given that the model is specifically trained on such patterns. However, it is unclear how this affects other instruction-following abilities — do they remain stable, or do they degrade? Can author demonstrate that?

**Questions:**

See my comments on the weakness

---

> ### Author Response · Authors · 2024-11-20
> **Author response to reviewer u3JS: Part (1)**
>
> We thank the reviewer for their comments regarding our “well written and easy to follow paper” introducing FIT. We hope this rebuttal response will address the reviewer’s comments regarding our introduction of Focus Instruction Tuning (FIT) in our paper.
>
> ---
>
> ## General Response regarding the focus of the introduction of our method FIT in this paper
> We respond to the individual points that the reviewer has highlighted, point by point:
> ### **G1: General Response.**
> Focus Instruction Tuning (FIT) is a general method that enables users to steer a model’s behaviour based on specified features [1,2,3,4]. This approach offers greater control over model outputs at inference, adding an important layer of explainability and controllability to model predictions.
>
> While the ability to understand and mitigate biases or spurious correlations is a valuable and natural application of FIT, it is not FIT’s only objective. Instead, the goal of steerability extends beyond bias mitigation to address broader challenges in managing and aligning model behaviour. For example, maintaining controllability is particularly important in contexts such as countering safety alignment fragility that can arise after fine-tuning [1]. In such cases, the ability to adapt model responses to align with user specifications is critical to ensuring safe and reliable deployment.
>
> We hope this clarification highlights the broader applicability and intent of FIT as a tool for model control and adaptability, with bias mitigation as a beneficial but secondary outcome. We will add further clarification of this to the final version of the paper to emphasise our main contributions and include further information regarding the steering of LLMs within the related work section of our paper.
>
> ---
>
> ## Our response to the individual points raised by the reviewer
>
>
> ### **P1: Training and evaluation in FIT require prior knowledge of spurious features.**
> We agree with the reviewer’s point, but the ability to incorporate domain expertise and adaptively specify pre-specified invariances can also be understood as a strength of our method in several scenarios:
>
> 1. **Alignment with Established Practices:**
>    Existing practices in industry and research [5,6] require identifying spurious correlations as a standard step in building robust and ethical machine learning systems. This pre-identification ensures that training is informed by an understanding of data biases and correlations, thereby preventing models from being trained and used "blindly."
>
> 2. **Regulatory and Ethical Expectations:**
>    Regulatory bodies and ethical guidelines are increasingly emphasising the need to identify and mitigate biases in AI systems [7]. Efforts are underway to define and enforce measurable categories of "violating behaviour" in models. As a result, it is becoming the responsibility of model developers to identify spurious features during development to ensure fair and unbiased predictions [8,9]. FIT is meant to complement this pre-identification process by providing a practical, adaptive tool for steering model behaviour at inference to understand and mitigate the presence of such identified features.
>
> 3. **Post-Deployment Mitigation:**
>    While pre-identification of spurious features is valuable, spurious correlations will become identifiable after deployment through real-world performance analysis. FIT supports iterative improvement, allowing developers to fine-tune models using focus instructions to address newly identified biases.
>
> 4. **Leveraging Automated Tools for Spurious Feature Detection:**
>    Moreover, pre-trained models can be used to generate focus labels automatically when explicit spurious feature labels are unavailable. For instance, our method is compatible with existing and future approaches for automatically identifying and labelling spurious features [12,13,14,15]. FIT can act as a complementary mechanism, providing steerability based on identified features, whether they are manually labelled or automatically detected.
>
> ---
>
> ### **P2: Narrow domain of improvement and model still relying on “focus and ignore instructions to control outputs.”**
> In **G1**, we clarify that the primary motivation behind FIT is to provide a flexible, adaptive method for steering model behaviour through dynamic focus and ignore instructions, rather than to solely mitigate spurious correlations. FIT’s applicability spans a wide range of tasks and feature complexities, and it retains the model’s general instruction-following capabilities (as demonstrated by the focus accuracy, corresponding to vanilla accuracy on empty focus instructions - ∅ in the figures).
>
> The adaptability of FIT is achieved through leveraging natural language focus instructions and feature specifications, allowing for a broad and versatile user-defined guidance that does not rely on rigid or pre-defined feature or focus instruction formats.

---

> > ### Author Response · Authors · 2024-11-29
> > **Response to Reviewer u3JS: Opportunity for Further Engagement**
> >
> > We appreciate the opportunity provided by the extended deadline to engage with the reviewer. We hope our responses have clarified that FIT is a general method for steering model behaviour, with applications that extend beyond bias mitigation, and that we have addressed all of the reviewer’s other comments.
> >
> > With the deadline for comments fast approaching on 2nd December, we look forward to hearing back from the reviewer and welcome the opportunity to address any further points or clarifications they may wish to raise.

---

> > ### Author Response · Authors · 2024-12-02
> > **Final Follow-Up: Response to Reviewer u3JS and Request for Feedback**
> >
> > Having not heard from the reviewer during the rebuttal period, and with the discussion period concluding today (2nd December), we would like to kindly follow up.
> >
> > We sincerely value your feedback and hope that our responses have satisfactorily addressed your comments and concerns. If this is the case, we would greatly appreciate it if you could reflect this in your final assessment.

---

> ### Author Response · Authors · 2024-11-20
> **Author response to reviewer u3JS: Part (2)**
>
> ### **P3: Spurious correlations originate from SFT data, diversity and quality in training data is needed.**
> We fully agree. FIT, like vanilla SFT, can benefit from high-quality and diverse datasets. Importantly, FIT serves as an orthogonal yet complementary approach for mitigating spurious correlations by providing a flexible mechanism to dynamically steer model behaviour during inference.
>
> This is achieved through focus and ignore instructions that are broad in scope, leveraging flexible natural language specifications and demonstrating applicability to general features of varying complexity, as shown across our SS, SMNLI, and BBQ experiments.
>
> The adaptability of FIT to work with new features and feature values, as demonstrated in our SMNLI and BBQ experiments (Figures 5 and 6), provides positive evidence that training across multiple tasks will be a valuable next step toward enhancing FIT’s flexibility and its generalisation to new features at inference.
>
> While FIT introduces the "focus and ignore" dimension, this complements, rather than replaces, efforts to improve the quality and diversity of SFT data along the other dimensions the reviewer mentions. This is indeed coherent with the current practice of combining multiple training instruction types to fine-tune LLMs. FIT represents an additional tool to add further variety and diversity.
>
> We will include an explicit discussion of this point in the final version of the paper.
>
> ---
>
> ### **P4: How are model’s instruction-following capabilities affected without focus instructions?**
> The empty focus instruction setting ($\emptyset$ in the figures) corresponds to the scenario in which the model's instruction-following capabilities are evaluated in the absence of explicit focus instructions.
>
> As shown in Figures 2, 3, 4, and 5, the demonstrated high focus accuracy for the empty focus instruction setting corresponds directly to the ground truth labels. This indicates that the instruction-following capabilities of LLMs are indeed preserved after applying FIT.
>
> We thank the reviewer for pointing this out and will incorporate additional clarification on this aspect in the paper.
>
> ---
>
> ## Conclusion and final comments
> We hope that our responses have adequately addressed the reviewer’s concerns. We remain available to respond to any further comments or questions the reviewer may have and welcome additional feedback.

---

> ### Author Response · Authors · 2024-11-20
> **Author response to reviewer u3JS: Part (3)**
>
> ## References
>
> [1] Bhattacharjee, A., Ghosh, S., Rebedea, T., & Parisien, C. (2024). Towards Inference-time Category-wise Safety Steering for Large Language Models. arXiv preprint arXiv:2410.01174.
> [2] Zhao, Y., Devoto, A., Hong, G., Du, X., Gema, A. P., Wang, H., ... & Minervini, P. (2024). Steering Knowledge Selection Behaviours in LLMs via SAE-Based Representation Engineering. arXiv preprint arXiv:2410.15999.
> [3] Panickssery, N., Gabrieli, N., Schulz, J., Tong, M., Hubinger, E., & Turner, A. M. (2023). Steering llama 2 via contrastive activation addition. arXiv preprint arXiv:2312.06681.
> [4] Li, K., Patel, O., Viégas, F., Pfister, H., & Wattenberg, M. (2024). Inference-time intervention: Eliciting truthful answers from a language model. Advances in Neural Information Processing Systems, 36.
> [5] OpenAI. (2024). Evaluating fairness in ChatGPT. OpenAI. Retrieved November 18, 2024, from https://openai.com/index/evaluating-fairness-in-chatgpt/
> [6] Microsoft. (2020, September 10). Diversity, inclusion, and responsible AI are now the bedrock of bias prevention. Microsoft. Retrieved November 18, 2024, from https://www.microsoft.com/en-us/industry/microsoft-in-business/business-transformation/2020/09/10/diversity-inclusion-and-responsible-ai-are-now-the-bedrock-of-bias-prevention/
> [7] Regulation, P. (2016). Regulation (EU) 2016/679 of the European Parliament and of the Council. Regulation (eu), 679, 2016.
> [8] Guldimann, P., Spiridonov, A., Staab, R., Jovanović, N., Vero, M., Vechev, V., ... & Vechev, M. (2024). COMPL-AI Framework: A Technical Interpretation and LLM Benchmarking Suite for the EU Artificial Intelligence Act. arXiv preprint arXiv:2410.07959.
> [9] Zeng, Y., Yang, Y., Zhou, A., Tan, J. Z., Tu, Y., Mai, Y., ... & Li, B. (2024). Air-bench 2024: A safety benchmark based on risk categories from regulations and policies. arXiv preprint arXiv:2407.17436.
> [10] Zhao, J., Wang, T., Yatskar, M., Ordonez, V., & Chang, K. W. (2018, June). Gender Bias in Coreference Resolution: Evaluation and Debiasing Methods. In Proceedings of the 2018 Conference of the North American Chapter of the Association for Computational Linguistics: Human Language Technologies, Volume 2 (Short Papers) (pp. 15-20).
> [11] Achiam, J., Adler, S., Agarwal, S., Ahmad, L., Akkaya, I., Aleman, F. L., ... & McGrew, B. (2023). GPT-4 technical report. arXiv preprint arXiv:2303.08774.
> [12] Wang, Z., & Culotta, A. (2020, November). Identifying Spurious Correlations for Robust Text Classification. In Findings of the Association for Computational Linguistics: EMNLP 2020 (pp. 3431-3440).
> [13] Zhou, Y., Xu, P., Liu, X., An, B., Ai, W., & Huang, F. (2023). Explore Spurious Correlations at the Concept Level in Language Models for Text Classification. arXiv preprint arXiv:2311.08648.
> [14] Wang, T., Sridhar, R., Yang, D., & Wang, X. (2022, July). Identifying and Mitigating Spurious Correlations for Improving Robustness in NLP Models. In Findings of the Association for Computational Linguistics: NAACL 2022 (pp. 1719-1729).
> [15] Zheng, G., Ye, W., & Zhang, A. (2024). Learning Robust Classifiers with Self-Guided Spurious Correlation Mitigation. arXiv preprint arXiv:2405.03649.

---

> ### Author Response · Authors · 2024-11-24
> **Follow up to rebuttal response to Reviewer u3JS**
>
> To promote a productive discussion period, we would greatly appreciate the reviewer's feedback on our response to their review. We are more than happy to provide any further clarification or engage in further discussion.

---

### Official Review · Reviewer_wRv3 · 2024-11-04

**Soundness:** 2
**Presentation:** 3
**Contribution:** 2
**Rating:** 6
**Confidence:** 3

**Summary:**

This paper introduces Focus Instruction Tuning, a finetuning strategy for focusing LLMs on specific features.  The method is applied to classification settings with spurious correlations.  Results shown on three tasks indicate for potential to mitigate bias over traditional finetuning.

**Strengths:**

- The proposed work targets improving robustness to spurious correlations. Improving robustness to spurious correlations is an important problem  that has been well studied in classification problems and is also important to study for more recent LLMs.
- The paper is overall well-written.  I appreciate the authors have highlighted the key takeaways one act experiment.

**Weaknesses:**

- The experiments for the paper consider older tasks including NLI and sentiment.  These datasets are old, and there are already many existing approaches for reducing bias in these datasets which the authors have not considered in this work.  I recommend the authors to compare against existing approaches for example: https://aclanthology.org/2020.acl-main.769/ and https://aclanthology.org/2022.acl-long.190.pdf.  Further the author could consider more recent benchmark tasks that the models are typically evaluated on as even the included QA dataset is a few years old (2022).
- The proposed method is limited to classification in the experiments, however there are also many tasks these LLMs are capable of including general QA, MCQ, generation, etc. which may have bias and also important to study.
- The evaluations that are done do not seems to be consistent with prior works as they separate based on the label for computing accuracy.  Could the authors also include the full accuracy on the datasets for comparison with prior work?

**Questions:**

- Can the authors clarify on the difference between FIT and IT? More specifically how is the $I_{focus}$ included in the data? Is this included for the IT comparison, or is the data left out? I believe including this data as part of $x$ would be a necessary comparison as well.

---

> ### Author Response · Authors · 2024-11-20
> **Author response to reviewer wRv3: Part (1)**
>
> We thank the reviewer for their comments regarding the introduction of our “simple, flexible and intuitive” method. We hope this rebuttal response will address the reviewer’s comments regarding our introduction of Focus Instruction Tuning (FIT) in our paper.
>
> ## General Response regarding the focus of the introduction of our method FIT in this paper
>
> ### **G1: General Response.**
> Focus Instruction Tuning (FIT) is a general method that enables users to steer a model’s behaviour based on specified features [1,2,3,4]. This approach offers greater control over model outputs at inference, adding an important layer of explainability and controllability to model predictions.
>
> While the ability to understand and mitigate biases or spurious correlations is a valuable and natural application of FIT, it is not FIT’s only objective. Instead, the goal of steerability extends beyond bias mitigation to address broader challenges in managing and aligning model behaviour. For example, maintaining controllability is particularly important in contexts such as countering safety alignment fragility that can arise after fine-tuning [1]. In such cases, the ability to adapt model responses to align with user specifications is critical to ensuring safe and reliable deployment.
>
> We hope this clarification highlights the broader applicability and intent of FIT as a tool for model control and adaptability, with bias mitigation as a beneficial but secondary outcome. We will add further clarification of this to the final version of the paper to emphasise our main contributions and include further information regarding the steering of LLMs within the related work section of our paper.
>
> ---
>
> ## Our response to the individual points raised by the reviewer
> We respond to the individual points that the reviewer has highlighted, point by point:
>
> ### **P1: existing techniques for removing biases for NLI and sentiment datasets.**
> As highlighted in **G1**, the primary objective of our work is to introduce FIT, a method for adaptively steering LLM generations at inference time using feature-specific focus instructions, rather than to evaluate FIT solely as a bias mitigation tool. Unlike the cited methods, which focus exclusively on bias mitigation (e.g., [5] incorporates an additional bias model - adding complexity - in the SFT training loss used for fine-tuning a separate language model, but lacks test-time adaptability), FIT enables flexible control and generalisation to unseen configurations or distribution shifts, as demonstrated in our SMNLI and BBQ experiments (see Sections 4.2 and 4.3). Therefore, direct comparison with these techniques is not appropriate, given their distinct focus on bias mitigation versus our emphasis on steerability. We will clarify this distinction further in the final version.
>
> ### **P2: experiments limited to classification, LLMs can do QA, MCQ, Generation and are important to study.**
> In our experiments, we consider both classification and **multiple-choice question-answering (MCQ)** (BBQ, Section 4.3). The lack of applications to open-ended NLG is not inherently related to a limitation of FIT, but a result of the cost and difficulty of producing high-quality NLG datasets. Indeed, applying FIT would have required us to perform the expensive and time-consuming process of generating high-quality focus targets to augment ground-truth targets that come with existing NLG datasets. Given that we are primarily concerned in this work with introducing a novel methodology, not contributing new datasets, we believe the endeavour of performing extensive and high-quality data collection to be out-of-scope but worth future exploration. Therefore, we will discuss applying FIT to open-ended NLG, and the associated data collection required, as an important direction for future work.

---

> ### Author Response · Authors · 2024-11-20
> **Author response to reviewer wRv3: Part (2)**
>
> ### **P3: Evaluations inconsistent with previous works, metric is different and separates based on label for computing accuracy.**
> Given **G1**, it should be clear that our goal is to study how the model’s behaviour changes based on specified focus instructions, not how good of a bias mitigation technique FIT is. The goal of focus accuracy is to measure the effectiveness at steering the behaviour of LLMs. This is in line with existing bias works that do not focus on raw accuracy, and instead report worst-group accuracy, accuracy gap and other more specific metrics beyond dataset-wide accuracy depending on the notion of fairness they aim to address [6,7,8].
>
> Standard accuracy alone would not highlight the behavioural differences induced by our model that we wish to investigate in this work. For example, a fine-tuned model might show high standard accuracy across all focus instruction types by defaulting to the ground-truth answer, regardless of what it is instructed to focus on or to ignore (e.g. to focus on a non-causal feature) - thus failing to exhibit steerability, which is the core motivation behind FIT. Hence, why we introduce and report focus accuracy which directly measures for test time controllability.
>
> For completeness, we will include standard accuracy for comparison in the appendix but as a specific example, we include the zero-shot (no fine-tuning using SFT or FIT and no few-shot prompting at inference) focus accuracies of the Llama.3.1-8B model on the BBQ dataset:
>
> | **Metric**                          | **Accuracy** |
> |-------------------------------------|--------------|
> | **Default Prompt Accuracy**         | 0.5939       |
> | **Focus on Causal Accuracy**        | 0.6242       |
> | **Focus on Spurious Accuracy**      | 0.3328       |
> | **Don't Focus on Spurious Accuracy**| 0.6377       |
> | **Focus on Causal, Don't Focus on Spurious Accuracy** | 0.6521 |
> | **Focus on Spurious, Don't Focus on Causal Accuracy** | 0.2779 |
>
> By comparing these results to the corresponding results in Figure 6 (middle column, bottom-row), we see that FIT sees improvements in causal accuracy (as shown, for example, via the improvements in default prompt accuracy) but also steerability (as shown by the improvements in focussing on non-causal features).

---

> ### Author Response · Authors · 2024-11-20
> **Author response to reviewer wRv3: Part (3)**
>
> ### **Q1: how is $I_{\text{focus}}$ included in the baselines.**
> Finally, we would like to clarify that focus specifications ($I_{\text{focus}}$) are included in our baselines during evaluation. We describe the experimental setup in Section 4 of the paper i.e., for the SFT baseline, we do not include focus instructions during training, and train on input-focus label pairs (this is identical to FIT but without the addition of the focus labels at training time). This gives a fair comparison between the SFT and baselines as they are trained on the same input-label pairs.
>
> As a further additional baseline within the paper, we also include a few-shot baseline using the pre-trained models with no SFT or FIT applied. When performing inference for a specific type of focus instruction (e.g. focus(C)), we randomly sample a context of 5 training examples, and append these as in-context examples using the same focus instruction (e.g. focus(C)) with the corresponding focus label.
>
> Moreover, in the appendix, we will also include vanilla SFT results (trained on input-ground truth pairs). As an example, the focus accuracies (on SMNLI) of the Mistral model are given in the table below:
>
> | **Metric**                          | **Same IID** | **Higher Correlation** | **Reduced Correlation** | **Flipped** |
> |-------------------------------------|--------------|-------------------------|--------------------------|-------------|
> | **Default Prompt Accuracy**         | 0.9067       | 0.8978                 | 0.8900                  | 0.8944      |
> | **Focus on Causal Accuracy**        | 0.8978       | 0.8933                 | 0.8911                  | 0.8933      |
> | **Focus on Spurious Accuracy**      | 0.3389       | 0.8200                 | 0.1489                  | 0.1022      |
> | **Don't Focus on Spurious Accuracy**| 0.9056       | 0.8911                 | 0.8878                  | 0.8822      |
> | **Focus on Causal, Don't Focus on Spurious Accuracy** | 0.8989 | 0.8922                 | 0.8878                  | 0.8889      |
> | **Focus on Spurious, Don't Focus on Causal Accuracy** | 0.3422 | 0.8189                 | 0.1489                  | 0.0956      |
>
> As we can see, comparing the results in this table to the results for the Mistral model presented in Figure 4 (first column, bottom row), FIT training retains the general instructions following capabilities and causal accuracy of traditional SFT (for example both SFT and FIT show 89.3% and 90.1% accuracy respectively on the default empty focus instruction corresponding to standard IT on the high-correlation subset).
>
> However, we can see FIT also makes the model clearly more steerable (as shown by the 98.0% vs 9.55% focus accuracy on the focus(S) and ignore(C) instruction on the flipped subset) - the core purpose behind our introduction of FIT.
>
> These trends continue across datasets and models and we will include the complete vanilla SFT results in the appendix of the final version of the paper.
>
> ---
>
> ## Conclusion and final comments
>
> We hope that our responses have adequately addressed the reviewer’s concerns. We remain available to respond to any further comments or questions the reviewer may have and welcome additional feedback.
>
> ---
>
> ## References
> [1] Bhattacharjee, A., Ghosh, S., Rebedea, T., & Parisien, C. (2024). Towards Inference-time Category-wise Safety Steering for Large Language Models. arXiv preprint arXiv:2410.01174.
> [2] Zhao, Y., Devoto, A., Hong, G., Du, X., Gema, A. P., Wang, H., ... & Minervini, P. (2024). Steering Knowledge Selection Behaviours in LLMs via SAE-Based Representation Engineering. arXiv preprint arXiv:2410.15999.
> [3] Panickssery, N., Gabrieli, N., Schulz, J., Tong, M., Hubinger, E., & Turner, A. M. (2023). Steering llama 2 via contrastive activation addition. arXiv preprint arXiv:2312.06681.
> [4] Li, K., Patel, O., Viégas, F., Pfister, H., & Wattenberg, M. (2024). Inference-time intervention: Eliciting truthful answers from a language model. Advances in Neural Information Processing Systems, 36.
> [5] Rabeeh Karimi Mahabadi, Yonatan Belinkov, and James Henderson. 2020. End-to-End Bias Mitigation by Modelling Biases in Corpora. In Proceedings of the 58th Annual Meeting of the Association for Computational Linguistics, pages 8706–8716, Online. Association for Computational Linguistics.
> [6] Sagawa, S., Koh, P. W., Hashimoto, T. B., & Liang, P. (2019). Distributionally robust neural networks for group shifts: On the importance of regularization for worst-case generalization. arXiv preprint arXiv:1911.08731.
> [7] Zhang, M., Sohoni, N. S., Zhang, H. R., Finn, C., & Ré, C. (2022). Correct-n-contrast: A contrastive approach for improving robustness to spurious correlations. arXiv preprint arXiv:2203.01517.
> [8] Wang, Z., & Culotta, A. (2020, November). Identifying Spurious Correlations for Robust Text Classification. In Findings of the Association for Computational Linguistics: EMNLP 2020 (pp. 3431-3440).

---

> ### Author Response · Authors · 2024-11-24
> **Follow up to rebuttal response to Reviewer wRv3**
>
> To promote a productive discussion period, we would greatly appreciate the reviewer's feedback on our response to their review. We are more than happy to provide any further clarification or engage in further discussion.

---

> ### Author Response · Authors · 2024-11-29
> **Response to Reviewer wRv3: Opportunity for Further Engagement**
>
> We appreciate the opportunity provided by the extended deadline to engage with the reviewer. We hope our responses have clarified that FIT is a general method for steering model behaviour, with applications that extend beyond bias mitigation, and that we have addressed all of the reviewer’s other comments.
>
> With the deadline for comments fast approaching on 2nd December, we look forward to hearing back from the reviewer and welcome the opportunity to address any further points or clarifications they may wish to raise.

---

> > ### Comment · Reviewer_wRv3 · 2024-12-01
> >
> > Thank you for addressing many of my comments and additional analysis.
> >
> > While I agree that the proposed method is more general and can be applied to many tasks and general steerability of LLMs, the evaluation is limited to classification (BBQ is very similar to classification as Reviewer duMo has also pointed out), and bias (or spurious correlations).  As such, I think it's important to compare with this line of work, or extend the proposed approach to other tasks/applications.
> >
> > My other concerns regarding I_{focus} and accuracy have mostly been addressed. I have adjusted my score accordingly based on these concerns.

---

> ### Author Response · Authors · 2024-12-03
> **Follow up to reviewer's final comment regarding comparisons with a debiasing technique**
>
> Thank you for your constructive feedback and suggestions. To address your concerns regarding comparisons with debiasing techniques, we have supplemented our work by implementing and adapting the product of experts (PoE) debiasing technique as suggested [5].
> We adapted the PoE method for the autoregressive models used in our paper by extracting and normalizing the logits of the first newly-generated token positions over the set of single tokens representing the answer options. We train the bias model on the stereotypical and biased response for each dataset example. Below, we present the focus accuracies for the Llama-8b-Instruct model on the BBQ dataset:
>
> | Metric                                              | Focus Accuracy     |
> |-----------------------------------------------------|--------------------|
> | Default Prompt Accuracy                             | 0.9944            |
> | Focus on Causal Accuracy                            | 0.9936            |
> | Focus on Spurious Accuracy                          | 0.25              |
> | Don't Focus on Spurious Accuracy                    | 0.9936            |
> | Focus on Causal, Don't Focus on Spurious Accuracy   | 0.9944            |
> | Focus on Spurious, Don't Focus on Causal Accuracy   | 0.25              |
>
> As shown, the PoE method performs comparably well—if not slightly worse—than FIT in debiasing (compare the standard focus(C) accuracy for FIT at 99.60% - c.f. Figure 6 - with the default prompt accuracy for PoE at 99.44%; both metrics correspond to causal accuracy for these prompt types). However, the PoE method requires training two separate models and does not provide steerability at test time as shown by the low focus accuracy on focus(S). Indeed the model defaults to the ground truth label across all prompt types and does not change behaviour despite different different focus specifications. This highlights the flexibility of FIT, which not only debiases effectively but also enables additional controllability during inference.
>
> We will include these results in the final version of the paper to address your suggestion and strengthen our comparison. We hope this analysis resolves your concerns regarding comparisons with debiasing techniques and that you will kindly consider adjusting your score to reflect this additional analysis.

---

### Author Response · Authors · 2024-12-04
**Summary of Rebuttal Period for Reviewers/AC/SAC to help with review process going forward.**

With the discussion period coming to an end, we provide a summary of the rebuttal and ensuing discussion to assist reviewers, AC, and SAC in the decision-making process. We have thoroughly addressed all initial comments and engaged in productive follow-ups with reviewers wRv3, cfLA, and duMo. While u3JS did not provide follow-up feedback, we hope their concerns were thoroughly addressed, as was the case with the majority of comments from the other reviewers.

Below are the key clarifications and contributions highlighted during the rebuttal:

**FIT as primarily a steering method**: FIT offers enhanced control over model outputs during inference, adding explainability and controllability. We provided a motivating example demonstrating its utility in post fine-tuning alignment. While mitigating biases or spurious correlations is a natural application, it is not FIT’s only objective.

**Instruction-Following Capabilities**: FIT preserves the instruction-following abilities of pre-trained language models while introducing focused adaptability.

**Novelty and Comparisons**:
- FIT is a novel methodology distinct from approaches like Context Distillation. While one application is mitigating spurious correlations, FIT uniquely focuses on refocusing model outputs with adaptable feature specifications at test time, making direct comparisons with traditional debiasing techniques less appropriate.
- Nonetheless, comparisons with an existing debiasing method demonstrates that FIT performs comparably or better while being computationally cheaper and more adaptable.
- Moreover, the novelty of our method and its distinction to the existing methods above justify the introduction of the new focus accuracy metric to evaluate FIT’s success in steering model behavior at inference time.

**Access to Spurious Features**: While FIT requires some information about spurious features during training, this is a modest requirement in-line with other related works and real-world practice. Furthermore, spurious feature identification methods can be easily deployed at inference-time to complement FIT by identifying spurious features on-the-fly.

**Dataset Choices**: Our focus on classification and multiple-choice QA datasets in our experiments stems from the cost and challenges of creating high-quality datasets for open-ended NLG training, not limitations of FIT itself. Introducing a novel methodology was prioritised over data collection, but future exploration is encouraged.

We hope that this proves useful to the reviewers/AC/SAC when evaluating the contribution of our work.

---

### Meta-Review · Area_Chair_iXXk · 2024-12-24

**Metareview:**

## Summary:
The paper introduces Focus Instruction Tuning (FIT), a method that enhances Instruction Tuning (IT) in training large language models (LLMs) by enabling them to prioritize specific features while disregarding others. FIT aims to guide LLMs to focus on task-causal features and avoid spurious correlations and biases, leading to improved model robustness, fairness, and explainability. This approach allows models to adaptively steer their behavior by focusing on different features during inference, facilitating more robust and fair LLM applications in real-world scenarios. Experimental results across three NLP tasks demonstrate FIT's effectiveness in managing spurious and causal features, mitigating biases, and generalizing to new contexts, indicating its potential to address issues related to biased or undesired model behaviors in diverse settings.

## Strengths:
1. FIT presents a simple, intuitive, and effective method of conditional finetuning that mitigates spurious correlations and biases, thus enhancing instruction-tuning's robustness.
1. The proposed condition can control the training process to ignore or focus on predefined features when predicting the labels.
1. Across three models and evaluations, FIT demonstrates effectiveness, showing improvement over the standard IT baseline.

## Weaknesses:
1. Despite the importance of the addressed problem of mitigating spurious correlations, the technical novelty of this paper is incremental: it is a straightforward application of conditional finetuning, which has been widely used to encourage/discourage features in LLM outputs under user-defined conditions. The main novelty is the design of the conditions (ignore, focus, ignore and focus) in Eq. 1-2 to mitigate biases.
1. Creating the conditions requires extra human efforts and prior knowledge regarding the bias features and their spurious correlation to the labels. Since these conditions provide extra supervision information to the training and inference, the improvement over non-conditional finetuning might not be that surprising.
1. It is not clear whether FIT affects the performance of LLMs on following other instructions: an evaluation of the FIT-trained model on standard instruction following benchmarks can tell whether the general capability degrades or not.
1. The experiments are limited to three easy and old tasks/datasets mainly focusing on classifications and multi-choice QA. Considering the limited diversity of samples in these datasets, it is not clear whether the test set performance can indicate better robustness to unseen similar spurious correlations. It is also not clear whether FIT will degrade the models' other capabilities without the evaluations on other standard benchmark tasks (which is a common risk of instruction tuning, especially when it focuses on a narrow domain). The authors provided a few additional experiments to expand the scope of the empirical study but they are not thorough enough to fully resolve the above concerns.

## Decision:
The authors provided further clarifications and additional experimental results in the rebuttal, as requested by the reviewers. Initially, only one reviewer supported the acceptance, but another reviewer who gave 5 raised the rating to 6 after confirming that the responses successfully addressed most of their concerns. However, the other two reviewers have not changed their original ratings (3 and 5), though one participated in the discussion. Given this scenario, the meta-reviewer carefully read the responses and the new experiments but found several important concerns (as listed in the weaknesses), especially the ones from the reviewers giving 3 and 5, that are not fully resolved. The authors' efforts in the short discussion period are appreciated but the new experiments are not thorough or still missing some key points and cannot fully address the original concerns. Several key points in the review comments are left for future works according to the authors' responses. Although the paper did demonstrate the proposed method's effectiveness on a specific type of task, because of the above, the overall contributions are still not enough for an ICLR publication. Moreover, it is not clear whether FIT trained on specific tasks degrades the general instruction-following capability. The authors are encouraged to keep strengthening the work and submit it to the next conference.

**Additional Comments On Reviewer Discussion:**

The authors provided further clarifications and additional experimental results in the rebuttal, as requested by the reviewers. Two reviewers giving positive ratings (8 and 6) confirmed that most of their concerns have been successfully addressed by the responses. The meta-reviewer carefully read the responses and the new experiments but found several important concerns (as listed in the weaknesses) of the two reviewers (which also overlap with the other two reviewers) are not fully resolved. Despite the efforts of authors in the short discussion period, the new experiments are not thorough or still missing some key points and cannot fully address the original concerns. Several key points in the review comments are left for future works according to the authors' responses.

---

### Decision · Program_Chairs · 2025-01-22

Reject